# Chemical multi-fingerprinting of exogenous ultrafine particles in human serum and pleural effusion

Dawei Lu [1,2], Qian Luo[3], Rui Chen[4], Yongxun Zhuansun[4], Jie Jiang[5], Weichao Wang[1,2], Xuezhi Yang[1,2], Luyao Zhang[1,2], Xiaolei Liu[1,2], Fang Li[3], Qian Liu [1,2,6✉] & Guibin Jiang[1,2✉]

Ambient particulate matter pollution is one of the leading causes of global disease burden. Epidemiological studies have revealed the connections between particulate exposure and cardiovascular and respiratory diseases. However, until now, the real species of ambient ultrafine particles (UFPs) in humans are still scarcely known. Here we report the discovery and characterization of exogenous nanoparticles (NPs) in human serum and pleural effusion (PE) samples collected from non-occupational subjects in a typical polluted region. We show the wide presence of NPs in human serum and PE samples with extreme diversity in chemical species, concentration, and morphology. Through chemical multi-fingerprinting (including elemental fingerprints, high-resolution structural fingerprints, and stable iron isotopic fingerprints) of NPs, we identify the sources of the NPs to be abiogenic, particularly, combustion-derived particulate emission. Our results provide evidence for the translocation of ambient UFPs into the human circulatory system, and also provide information for understanding their systemic health effects.

[1] State Key Laboratory of Environmental Chemistry and Ecotoxicology, Research Center for Eco-Environmental Sciences, Chinese Academy of Sciences, Beijing 100085, China. [2] College of Resources and Environment, University of Chinese Academy of Sciences, Beijing 100190, China. [3] Shenzhen Institutes of Advanced Technology, Chinese Academy of Sciences, Shenzhen 518055, China. [4] Sun Yat-sen Memorial Hospital of Sun Yat-sen University, Guangzhou 510120, China. [5] Shenzhen Center for Disease Control and Prevention, Shenzhen 518055, China. [6] Institute of Environment and Health, Jianghan University, Wuhan 430056, China. ✉email: qianliu@rcees.ac.cn; gbjiang@rcees.ac.cn

Although humans have been exposed to ambient particulate matter throughout their history, only in recent decades did it become one of the leading global health risks owing to dramatically increased anthropogenic sources[1]. The rapid development of nanotechnology is likely to raise another source through inhalation, ingestion, skin contact, and injection of engineered nanoparticles (NPs)[2]. Ambient fine particulate matter (PM$_{2.5}$) with a size of <2.5 μm is of prior concern because it can penetrate deep into human bronchus and lungs. More than 90% of the global population (about seven billion people) are living in polluted air with PM$_{2.5}$ exceeding the guideline limit (10 μg m$^{-3}$) set by the World Health Organization (WHO)[3]. In 2015, PM$_{2.5}$ was ranked as the fifth global risk factor that caused 7.6% of total global mortality[1]. Long-term exposure to PM$_{2.5}$ is thought to increase mortality and morbidity and shorten life expectancy by causing cardiovascular and respiratory diseases, such as respiratory infections, chronic obstructive pulmonary (COPD), heart attack, stroke, and lung cancer[4,5].

When considering the health risks of particulate matter, a pivotal point is that the smaller the particulate matter is, the deeper it can enter into human body. Particulate matter may go through olfactory nerve to get access to human brain[6,7]. Moreover, it is generally thought that ambient ultrafine particles (UFPs) with a size of <0.1 μm are able to enter pulmonary alveoli, go through their walls, and translocate into the blood circulatory system[2,8,9]. Translocation into the blood circulation provides a mechanism for the cardiovascular effects associated with inhaled UFPs, and also suggests that inhaled UFPs can distribute throughout the human body to induce a systemic health effect beyond causing cardiovascular and respiratory diseases[10,11]. The translocation of inhaled UFPs has been comprehensively shown to occur in animals[12–15]. However, data in humans for the translocation of ambient UFPs into the blood circulation remains limited[16–18]. Recently, Miller et al.[17] reported that inhaled gold NPs can translocate from the lung to the circulation and accumulate at sites of vascular inflammation in human, which provided evidence for such translocation. Calderón-Garcidueñas et al.[18] reported that exogenous magnetic NPs were found in the human hearts, also suggesting the transport of NPs via the blood circulation. Despite that, the real occurrence states, chemical species, or sources of atmospheric UFPs in humans are scarcely known. Therefore, a crucial link in understanding the health effects of particulate pollution is still to be established.

Here, we report the direct probing and identification of exogenous NPs in non-occupational human serum and pleural effusion (PE). To open up the "black box" state of UFPs in the human circulatory system, we extract and capture NPs from non-occupational human serum and PE samples (n = 37). The samples were collected in a typical polluted region (Pearl River Delta, China) where the annual mean PM$_{2.5}$ concentration in 2016 was 32 μg mL$^{-1}$ [19] and the notable pollution sources include industrial coal combustion, vehicle emission, road dust, construction dust, and secondary inorganic aerosol[20]. At first, we attempt to investigate NPs in serum to directly reflect the internal exposure to UFPs. However, we find that the NP levels in serum are too low to perform a comprehensive characterization. Internal exposure of NPs in serum does not provide information on any dose-exposure relationship. Therefore, we also select PE as a target sample based on the following consideration: it has been reported that a fraction of inhaled particles deposited in the lung can reach the pleural cavity and drain via the stomata in the parietal pleura to pulmonary and mediastinal lymph nodes[21,22], which is known as a clearance mechanism via lymphatic pathway of inhaled particles[21]. This means that PE should also contain an amount of inhaled UFPs. Thus, we hypothesize that PE can serve as a potential host of inhaled UFPs in the human body.

The NPs extracted from serum and PE samples are purified by enzymatic hydrolysis and chemical digestion to remove the adsorbed biological ingredients (see Methods), and then the chemical multi-fingerprints (including elemental, structural, and stable isotopic fingerprints) of NPs are characterized. It should be noted that PE samples can only be clinically collected from patients with some specific diseases. So, in this study, both patients and healthy subjects are recruited, and serum samples are collected from both healthy subjects and patients, which allows a comparison between healthy subjects and patients, whereas PE samples are only from patients. The information for all study participants are given in Supplementary Table 1–3, and the serum and PE samples are numbered with initials S and P, respectively.

## Results

**Concentration and size distribution of NPs in human serum.** Nanoparticle tracking analysis (NTA) allows a rapid determination of size distribution profile of NPs in liquid. As shown in Fig. 1a, NTA shows that NPs are widely present in all of the human serum samples with the concentration ranging from $1.4 \times 10^8$ to $1.0 \times 10^{10}$ particles mL$^{-1}$ (mean value $2.6 \times 10^9$ particles mL$^{-1}$). The low-particle concentration in serum may result from the rapid clearance of NPs in blood by the mononuclear phagocyte system. The size distribution shows large polydispersity and individual differences within a few to hundreds of nanometers. Notably, the NP levels between healthy subjects and patients show no significant difference (P = 0.11, unpaired Student's two-tailed t test; Supplementary Fig. 1a), indicating that the presence of NPs in the circulatory system is common in the general population. It should be noted that NTA only gives a rough estimate of populations of NPs with a working range for particle size of 10–2000 nm and do not give any information on sources. Furthermore, NPs may undergo agglomeration or deagglomeration in body fluids or in extraction process and may also partially dissolve during the translocation process, which may strongly affect the particle concentration and size distribution. The biotransformation of NPs in the lung and tissues are also possible. Thus, it should be aware that the particle size distributions obtained here may not exactly reflect those in ambient environment.

**Concentration and size distribution of NPs in human PE.** Abundant NPs are also observed in all of the PE samples (Fig. 1b and Supplementary Movie 1). Compared with those in serum, NPs in PE show two differences: (i) the particle concentration in PE ($4.2 \times 10^8$–$4.0 \times 10^{10}$ particles mL$^{-1}$) is much higher than that in serum (P < 0.001, unpaired Student's two-tailed t test; Fig. 1c). This supports the hypothesis that inhaled UFPs can transit directly to the pleural cavity and PE is a potential reservoir for inhaled UFPs, enabling the characterization of chemical multi-fingerprints of NPs. A previous paper also reported that NPs were present in the PE of occupational workers who had exposed to the NPs with severe lung damage[23], although the link of the NPs with the disease was unclear[24]. (ii) The NPs in PE are less polydisperse than those in serum (nearly all of the NPs in PE are constrained within 50–200 nm). The explanation for that is not clear but may be related to the dynamics (e.g., agglomeration/deagglomeration, dissolution, and transformations) of NPs in the body fluids and the "accumulation effect" when NPs enter or drain from PE[25]. The NP concentration in PE shows no significant correlation with age (Fig. 1d) or sex (P > 0.05, unpaired Student's two-tailed t test; Supplementary Fig. 2), and no significant differences in the NP levels in PE are found among different diseases (P > 0.2, unpaired Student's two-tailed t test; Supplementary Fig. 1b). Since the objective of the present study is to probe exogenous UFPs in the

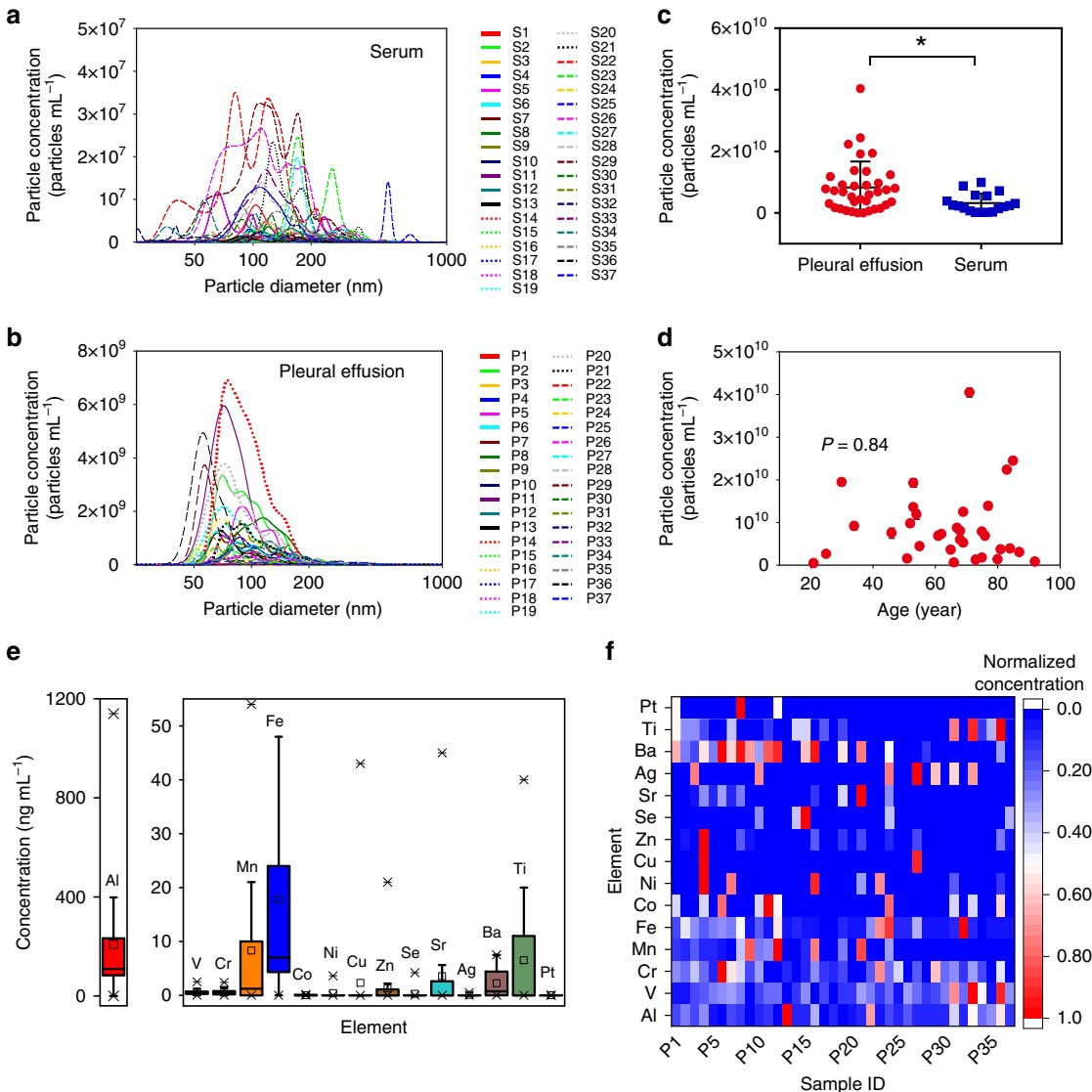

**Fig. 1 Concentration, size distribution, biological effect, and elemental fingerprints of NPs in human serum and pleural effusion. a** Concentration and size distribution of NPs in human serum samples. **b** Concentration and size distribution of NPs in human PE samples. **c** Comparison of particle concentrations in serum and PE. *$P = 0.0002$ (unpaired Student's two-tailed $t$ test). Error bars represent mean ± s.d. ($n = 37$). **d** Correlation of particle concentration in PE with the age. $P = 0.84$ (F test in linear regression analysis; $n = 37$). **e–f** Elemental concentration ranges **e** and heating map **f** of normalized elemental fingerprints of the NPs extracted from 37 human PE samples ($n = 37$). In **e** bounds of the box spans from 25% to 75% percentile, center line represents median, and whiskers visualize 5% and 95% of the data points. The high-abundance elements in organisms (e.g., C, H, O, N, S, P, Na, K, Ca, Mg, and Cl) are not shown. Source data are provided as a Source Data file.

human body, we herein do not study the correlation of the NPs with diseases. For reference, we perform a preliminary pro-inflammatory effect test for the PE-derived NPs (see Supplementary Fig. 3).

**Elemental fingerprints of NPs in PE.** To identify the sources of the NPs, we first characterized the elemental fingerprints of the PE-derived NPs. Overall, the NPs in PE show an extreme diversity in elemental fingerprints. Diverse transition metal and metalloid elements are detected in the NPs (Fig. 1e and Supplementary Table 4), including those are thought not to be essential in humans (e.g., Al, Ti, Ba, Sr, Ag, and Pt). Note that Al, Ti, Mn, Ba, and Sr, which are rock-forming elements or relatively abundant elements in the Earth's crust, show relatively high concentrations in the NPs (Fig. 1e), suggesting that the NPs are of external abiogenic sources. Remarkably, Pt, a characteristic element in catalysts[26], is found in some samples from benign

patients (with no application of Pt-containing agents). The elemental fingerprints of NPs also show large individual differences (Fig. 1f). In some samples (e.g., P18, P20, and P28), all elements of interest show very low concentrations, which may reflect low ambient particulate environment the subjects lived in.

**Structural fingerprints with enhanced chemical identities of NPs in PE.** The structural fingerprints of the PE-derived NPs are then uncovered by high-angle annular dark-field scanning transmission electron microscopy (HAADF-STEM), energy dispersive X ray spectroscopy (EDXS), and electron energy loss spectroscopy (EELS). HAADF-STEM with EDXS and EELS provides a powerful tool to obtain morphological and crystallographic information of NPs at atomic scale with enhanced chemical identities such as electronic structure and chemical bonding. Generally, we can observe diverse morphologies of NPs in a sample; meanwhile, NPs with similar morphologies can be

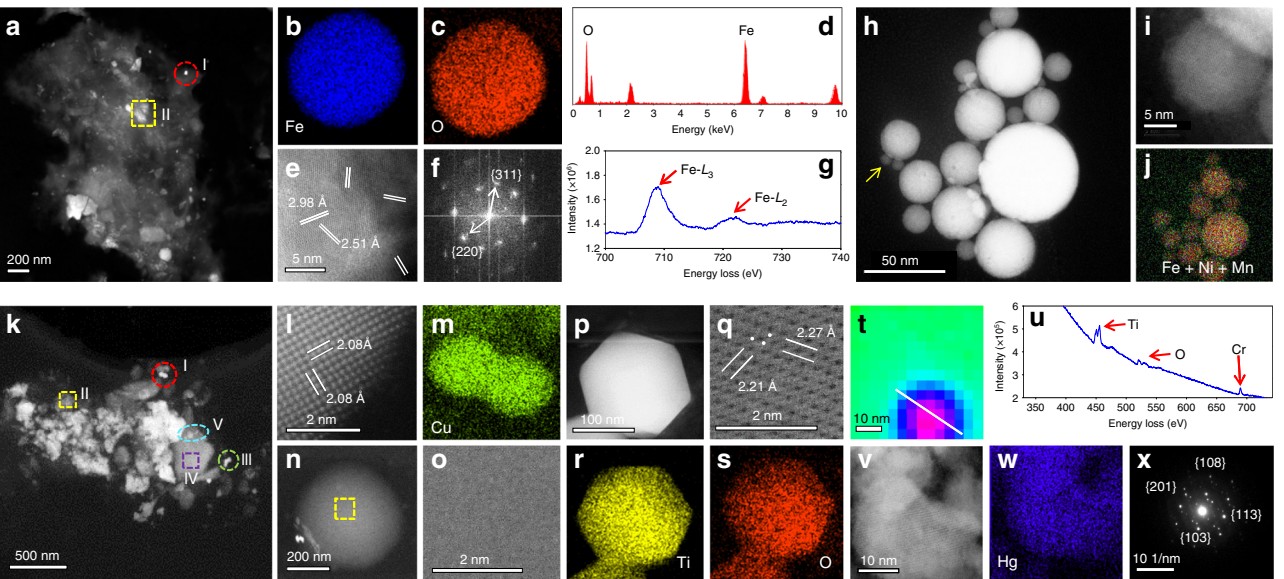

**Fig. 2 Structural fingerprints of typical NPs extracted from human PE samples. a** HAADF-STEM image of the NPs extracted from the P4 sample. **b–g** Characterization of a spherical particle marked in the red circle in **a** (section I). **b–c** EDXS mapping, **d** EDXS spectrum, **e** atomic resolution HAADF-STEM image, **f** fast Fourier transform (FFT) image, and **g** EELS spectrum of the particle. **h–j** Characterization of the particles marked in the yellow square in **a** (section II). **h** HAADF-STEM image and **j** EDXS mapping of the particles in the section II in **a**. **i**, High-resolution HAADF-STEM image of a round particle marked by the yellow arrow in **h**. **k**, HAADF-STEM image of the particles extracted from the P27 sample. **l** Atomic resolution HAADF-STEM image and **m** EDXS mapping of particles marked in the red circle in **k** (section I). **n** HAADF-STEM image and **o**. High-resolution HAADF-STEM image of a spherical particle marked in the yellow square in **k** (section II). **p** HAADF-STEM image, **q** atomic resolution HAADF-STEM image, and **r–s** EDXS mapping of hexagonal particles marked in the green circle in **k** (section III). **t** EELS mapping of particles marked in the purple square in **k** (section IV) showing a core-shell structure of the particles. **u** EELS spectrum of the core part of the particle in **t**. **v** High-resolution HAADF-STEM image, **w** EDXS mapping, and **x** Selected area electron diffraction (SAED) pattern of the particles marked in cyan ellipse in **k** (section V). Source data for **d**, **g**, **u** are provided as a Source Data file.

repeatedly observed in different samples. Figure 2 shows some frequently observed NPs in PE. Abundant Fe-bearing NPs are found in HAADF-STEM (the NPs-derived Fe makes up 0.017–1.23% of the total Fe in the PE; see Supplementary Table 4), with a large number of NPs consisting of Fe and O (section I in Fig. 2a; Fig. 2b–d). Indexing of the lattice fringes and the FFT pattern of the particles are consistent with the magnetite crystal structure (Fig. 2e–f). In EELS (Fig. 2g), the Fe-$L_3$ edge absorption at 709.1 eV and broad peak of the Fe-$L_2$ edge absorption, together with the integrated areas of the $L_3/L_2$ (~5.5) and the Fe/O ratio (0.81), also demonstrate that the particle is magnetite. Recall that endogenous magnetite particles can form via in vivo crystallization within the 8-nm-diameter core of ferritin[7,27,28]. The particle size of the magnetite found here greatly exceeds that of the ferritin-derived NPs. Furthermore, the magnetite NPs in PE show fused interlocking surface crystallites (noting the varying lattice orientations of the individual crystallite faces; see Supplementary Fig. 4 for higher resolution image), which is typical of high-temperature formation and subsequent crystallization upon rapid cooling and/or oxidation[7,18]. Thus, the magnetite NPs observed here contrast with the biogenic ones but highly resemble the pollution NPs found in the human brain and heart reported previously[7,18]. Magnetite particles can be produced by combustion of Fe-bearing organic substances or frictional heating processes (e.g., vehicle braking)[29–32]. Besides magnetite, we also observe other Fe-bearing NPs such as rounded crystal NPs only consisting of Fe, Mn, and Ni (Fig. 2h–j and Supplementary Fig. 5) that resemble Fe-Mn-Ni alloy particles existing in PM2.5 emitted from ferroalloy plants[33,34].

The Fe-bearing NPs co-occur with particles with diverse chemical compositions in PE (Fig. 2k). For example, Cu NPs are identified with a particle size of ~15 nm (Fig. 2l–m). Elemental Cu

is scarcely present in natural environment or human body, and it is most likely to be emitted by electric motors in indoor environments[35,36] or released from lumber because of the wide use of Cu as a wood preservative[37]. Amorphous spherical NPs with a characteristic elemental fingerprint of fly ash (Si, O, and C; see Supplementary Fig. 6), a frequent content in PM2.5 arising from the coal combustion, are widely present in PE (Fig. 2n–o). We also find hexagonal Ti-bearing particles (Fig. 2p). Lattice indexing and EDXS mapping indicate that they are crystal TiO2 NPs (Fig. 2q–s). Nano TiO2 is one of the most produced engineered nanomaterials and is often spherical in nature or occurs as nanorods, which have a hexagonal crystal lattice[38]. Some NPs show a heterogeneous core-shell structure consisting of multiple pollution elements (Fig. 2t–u). We even observe a few Hg-bearing crystal NPs, which are identified as HgS by lattice indexing and SAED pattern (Fig. 2v–x and Supplementary Fig. 7)[39,40]. Hg and its compounds are highly toxic, and fly ash is normally thought as the major carrier of Hg in particulate matter[41].

**Structural fingerprints of NPs in human serum.** We also endeavor to characterize the structural fingerprints of NPs in human serum. The much lower concentration of NPs in serum makes their characterization much more difficult than in PE. Despite that, diverse NPs are found with similar structural fingerprints with those in PE. As shown in Fig. 3a–f, particles consisting of Fe and O are widely present in serum samples from both healthy subjects and patients (Fig. 3b–c and e–f). Lattice indexing of the NPs fully match the crystal magnetite (Supplementary Fig. 8). Furthermore, as those found in PE, the magnetite NPs in serum also bear fused interlocking surface crystallites associating with high-temperature formation (Supplementary

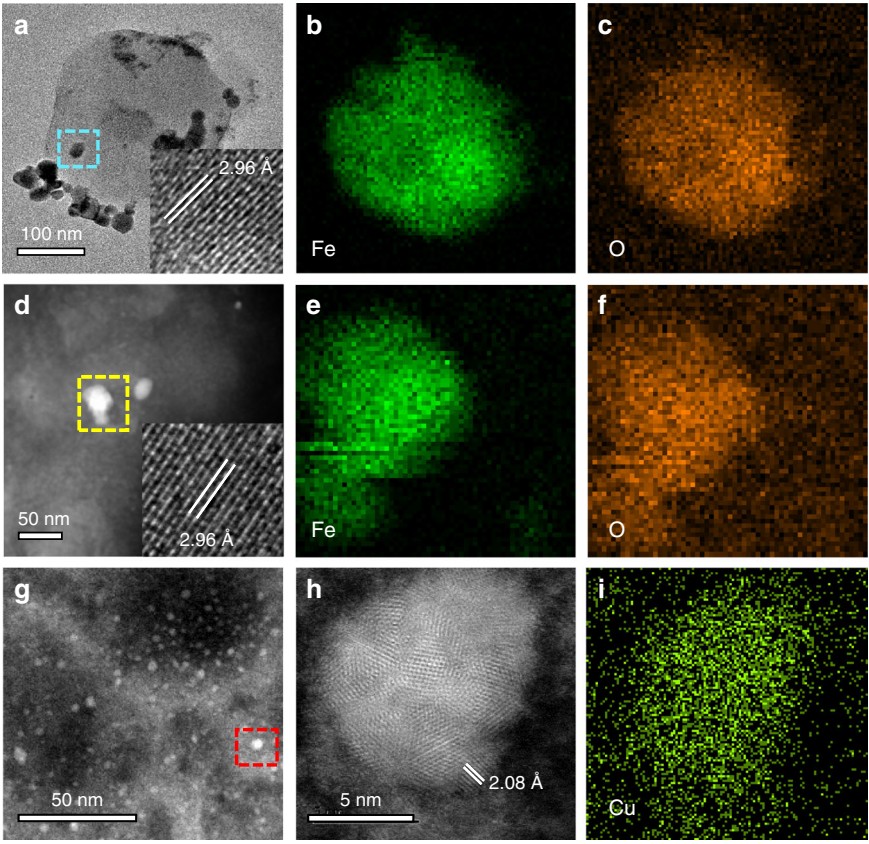

**Fig. 3 Structural fingerprints of typical NPs extracted from human serum samples. a** TEM image of the particles from a healthy human serum sample. The inset shows the high-resolution image of the selected section. **b–c** EDXS mapping of the particles marked in the cyan square in **a**. **d** HAADF-STEM image of the particles from a patient serum sample (S4). **e–f** EDXS mapping of the particles marked in the yellow square in **d**. **g** HAADF-STEM image of the particles from another patient serum sample (S27). **h** High-resolution HAADF-STEM image and **i** the corresponding EDXS mapping of the particles marked in the red square in **g**.

Fig. 8) and much larger particle size than endogenous ferritin-derived NPs (Supplementary Fig. 9). Cu NPs that have consistent shapes and lattice fringes with those in PE are also found (Fig. 3g–i). In addition, we identified a type of special spindle-shaped NPs with a mineral-like elemental composition (C, O, Ca, and P) in both serum and PE samples (Supplementary Fig. 10). The elemental composition may be suggestive of calcium carbonate or calcium phosphate, which can have both exogenic or endogenic origins. Overall, NPs in serum also show a large diversity (Supplementary Fig. 9), and, more importantly, the NPs in serum have their counterparts in PE with consistent structural fingerprints. The elemental and structural fingerprints of the NPs in serum and PE bear a high resemblance to air pollution particulates. Thus, it is rational to infer that a majority of the NPs in PE and serum are of same origin from ambient UFPs.

**Stable Fe isotopic fingerprints of NPs.** To further clarify the sources of NPs, we analyze the stable Fe isotopic fingerprints of the PE-derived NPs and the residual PE without NPs by using multicollector inductively coupled plasma mass spectrometry (MC-ICP-MS). Stable isotopic fingerprints of elements can record the information on the sources of NPs[42,43]. The PE-derived NPs and the residual PE were fully separated by repeated high-speed centrifugation at $106,21 \times g$. As shown in Fig. 4, the $\delta^{56}Fe$ values of the extracted NPs are in the range of −1.63–0.24‰ (mean value = −0.70 ± 0.55‰, $n = 14$) with no mass-independent fractionation (Supplementary Fig. 11). Significant differences in $\delta^{56}Fe$ between the NPs and the residual PE without NPs are

observed ($P < 0.05$, unpaired Student's two-tailed $t$ test; except only for P23). Nine samples of NPs are enriched in the light Fe isotopes, whereas four samples are enriched in the heavy Fe isotopes relative to the residual PE ($\Delta^{56}Fe$ up to 0.92‰). It is known that Fe isotopes are distributed heterogeneously in the human body, and each individual subject bears a certain Fe isotopic fingerprint in blood which is stable over long-term[44,45]. Therefore, the largely varied $\Delta^{56}Fe$ between NPs and PE indicates the external sources of the NPs.

Notably, most of the NPs show much negative $\delta^{56}Fe$ values than the crustal material (0.03‰)[46] but in line with anthropogenic fine particles emitted through combustion processes (e.g., vehicle exhausts, municipal solid waste incineration, industrial processes, and frictional heating of brake pads[31]), which can produce particles strongly depleted in the heavy Fe isotopes[47]. Therefore, it is suggested that combustion-derived particles may contribute considerably to the NPs. Note that biogenic materials, albeit also bearing isotopically light Fe[44,48], have been removed from the extracted NPs (see Methods), so they should not account for the low $\delta^{56}Fe$ values of the NPs. Only for a few samples (i.e., P5, P8, and P22), the $\delta^{56}Fe$ values are close to the crustal value, indicating that the NPs in these samples may be more associated with non-combustion sources (e.g., natural dust). Overall, Fe isotopic fingerprints support the result obtained through the elemental and structural fingerprinting that exogenous, particularly, combustion-derived particles may dominate the NPs in PE. This result is consistent with the air pollution characteristics in the studied region[20].

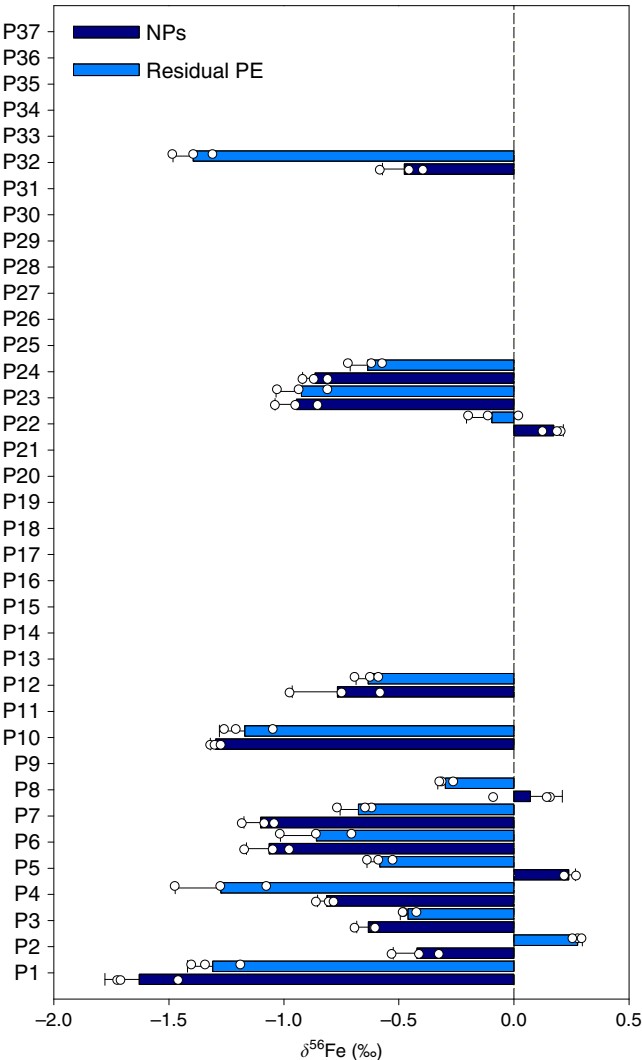

**Fig. 4 Stable Fe isotopic fingerprints of human PE samples and the NPs extracted from the PE samples.** The dark blue and light blue bars represent the NPs extracted from the PE samples and the corresponding residual PE samples without NPs, respectively. The error bars represent 1 s.d. from the isotopic ratio measurements ($n = 3$). The samples with no data mean that the Fe concentrations in these samples are too low to perform the isotopic analysis. Source data are provided as a Source Data file.

## Discussion

Our results have revealed the wide presence of exogenous NPs in non-occupational human PE and serum with extreme diversity in concentrations, chemical identities, and sources. The elemental, structural, and stable isotopic fingerprints suggest that the NPs are most likely to originate from external particulate pollution, particularly, combustion-related particulate emission. Although some NPs still keep the possibility of deriving from ingested food or water (or even the skin), our results suggest that pulmonary inhalation of ambient UFPs is a more likely exposure route. Until now, UFPs have not been classified as a criteria pollutant in the National Ambient Air Quality Standard[49] mainly owing to practical limitations of current environmental measurement. This preliminary study provides evidence in human for the systemic health effects of ambient UFPs. Furthermore, it is commonly supposed that ambient UFPs (<0.1 μm) are capable of penetrating through the pulmonary alveoli and going into the blood

cirsulation[2,8,9], although there is still some doubt as to whether particles in the range of 30–100 nm are too big to translocate given the known biological barriers, pore sizes, and translocation mechanisms. In this study, some particles extracted from serum and PE actually significantly exceed this threshold. The translocation of large particles (e.g., 240 nm) via inhalation pathway has also been reported to occur in animals[50]. Considering the potential agglomeration tendency of NPs in body fluids (and in extraction process), it may be premature to conclude that the penetrability of ambient particulate matter is stronger than previously thought. However, our results call for more-comprehensive studies on size-dependent health effect of particulate exposure. In addition, the methodology developed here, i.e., chemical multi-fingerprinting, can link the internal NP exposure to the external ambient UFPs, which adds a tool into the toolbox of exposomics studies and nanomaterial risk assessment.

So far, the following questions are still to be answered: (1) although we have identified different species of NPs, their contributions to the total toxicity of UFPs need further clarification, which also needs to take into account the effect of protein corona of the NPs and their solubility after uptake into human body in the future studies. (2) The estimates of NPs only represent a "snap-shot" of particle movement in the human body, and organ accumulation may be different for different types of NPs. The NPs in serum and PE will have a residence time, which can affect particle transformation, solubility, and clearance. Thus, the occurrence states of exogenous NPs in more human organs need to be studied to fully understand the dynamics and life cycle of ambient UFPs in the human body. (3) Owing to the limitations of the techniques used and their specific requirements for samples, a large proportion of elements or components (e.g., carbon, nitrogen, sulfate) of the inhaled UFPs are not addressed in the present study. The insoluble portion of these components may also translocate into the circulatory system in forms of particulates and drive toxic effects. Investigation on this portion of components needs integration of more techniques into the toolbox. (4) The connections between the presence of NPs in the human body and disease development are still unclear, and future studies will be needed under more pollution conditions and in more regions where the internal NP exposure in human body may be largely varied.

## Methods

**Study participants.** All human body fluids samples of patients (including serum and PE) were collected in Sun Yat-sen Memorial Hospital of Sun Yat-sen University (Guangzhou, Guangdong Province, China) during December 2015 and December 2016. Serum samples included 19 healthy individuals and 18 patients with lung diseases ($n = 37$). The serum samples of healthy individuals were collected from Shenzhen Center for Disease Control and Prevention (Shenzhen, Guangdong Province, China). The PE samples were collected from 37 patients with lung cancer, pneumonia, tuberculosis, heart failure, breast cancer, COPD, or POEMS syndrome ($n = 37$). To avoid any contamination, all sample collection was conducted in biological laminar air flow wards using ultra-clean disposable devices. After collection, the samples were immediately tightly sealed until analysis. The demographics and characteristics for study participants, including ages, genders, and history are provided in Supplementary Table 1–3. The healthy individuals presented no clinical evidence of diseases. The patients were classified by clinical diagnose. All participants denied any history of occupational exposure to hazardous materials. All participants provided informed consent and the study protocol was approved by the Ethics Committee of Shenzhen Institutes of Advanced Technology of Chinese Academy of Science and compliant with all relevant ethical regulations for studies involving human subjects.

**Extraction and purification of NPs from human body fluid samples.** To preclude any contamination and operator bias, all samples were handled in a laminar flow clean bench environment in an ultra-clean room. Ultrapure solvents were used through the experiments and all devices and containers were thoroughly washed with particle-free sterile water for several times before use. First, 1.5 mL of serum samples and 80 mL of PE samples were preconcentrated to 0.3 and 1.5 mL, respectively, by using high-speed centrifugation at 12851 × $g$ for 30 min. The

concentrated samples were then subjected to enzymatic hydrolysis and chemical digestion to remove adsorbed biological ingredients. To degrade the nucleic acids, the concentrated samples were mixed with particle-free benzonase (250 U mL$^{-1}$) at a ratio of 25:1 for 30 min at room temperature[51]. Afterwards, the samples with benzonase were digested by the addition of proteinase K (10 mg mL$^{-1}$) at a ratio of 2500:1 for 12 h at 55 °C. To ensure a complete digestion, the samples with proteinase K were further digested with TMAH (20%, v/v) for 48 h at room temperature at a ratio of 1:25[52]. The obtained solutions were purified by using centrifugal filter devices containing porous cellulose membranes (MWCO: 3 KD) with particle-free sterile water. This purification step was repeated three times to completely remove TMAH. Finally, the residual solids were solved with sterile water to a certain volume (serum: 0.5 mL; PE: 2 mL) and then stored in the dark at 4 °C. The whole extraction and purification procedures were validated by using standard addition method with a silica nanoparticle standard reference material (JEA0293). The recoveries were in the range of 76.7–86.2%, demonstrating the high effectiveness and efficiency of the method. The control experiments with solvents only were also conducted to ensure no contamination during the sample preparation procedures (particle number below detection limit in NTA and no pollution particulates were found under electron microscopes).

**Characterization of NPs.** The particle concentrations and size distribution were characterized by using a Nanosight NS300 Nanoparticle Tracking Analyzer (Malvern, UK). The accuracy of the measurement in serum and PE was calibrated with two silica nanoparticle standard reference materials, JEA0293 and JEA0232-ECP1433 (NanoComposix, San Diego, CA, USA). The working ranges of Nanosight NS300 for particle concentration and size are $10^7$–$10^9$ particles mL$^{-1}$ and 10–2000 nm, respectively. Prior to NTA measurement, a 0.05% (v/v) TWEEN-20 as a stabilizer was added to sample solution at a ratio of 1:25 (below the critical micelle concentration of TWEEN-20) to maintain the stability of NPs[53]. The morphology and surface textures of particles were characterized a Hitachi S-3000N scanning electron microscope (Tokyo, Japan), a JEM-ARM200F NEOARM atomic resolution analytical electron microscope (Tokyo, Japan), and a FEI Tecnai F20 field emission scanning transmission electron microscope (Hillsboro, USA). All electron microscopes were equipped with energy dispersive X ray spectroscopes, and JEM-ARM200F and FEI Tecnai F20 were also equipped with electron energy loss spectroscopies.

**Measurement of elemental concentrations.** The elemental concentrations (Al, V, Cr, Mn, Fe, Co, Ni, Cu, Zn, As, Se, Sr, Ag, Cd, Ba, U, Ti, and Pt) of the extracted particles and body fluids samples (serum and PE) were determined by an Agilent 8800 inductively coupled plasma mass spectrometer (Santa Clara, CA, USA). The samples were digested with concentrated HNO$_3$ using a CEM MARS 5 microwave sample preparation system (Matthews, NC, USA)[43]. In brief, 5 mL of 65% (v/v) HNO$_3$ and 1 mL of 35% (v/v) H$_2$O$_2$ were added to the sample, and then the mixture was irradiated at 180 °C (1600 W) for 30 min and 120 °C (800 W) for 10 min. After cooling, 200 μL of the digests were transferred to a 5 mL polyethylene centrifuge tube and then diluted to 4 mL with ultrapure water for ICP-MS analysis. To subtract the background signals from the results, the blank control sample was also analyzed following the same procedures. With the standard addition method, the recoveries of the elements ranged from 91.2% to 103.2%.

**Measurement of Fe isotopic ratios.** For isotopic analysis, the samples were digested as described above. The digests were evaporated to dryness and then diluted to 5 mL with 8 M HCl/0.001% H$_2$O$_2$. The Fe in samples was isolated and purified from the digests with an anion exchange chromatographic method[54]. In brief, a strong anion resin (AG MP-1, 100–200 mesh) was activated by mixing with ultrapure water at a ratio of 3:4 for 12 h and then filled to a 2 mL resin bed of a Bio-Rad Poly-Prep column. The resin was cleaned and conditioned with 0.7 M HNO$_3$, water, and 8 M HCl/0.001% H$_2$O$_2$ (v/v) in sequence. After sample loading, the matrix and Fe could be sequentially eluted with different eluents. In order to remove the interference of the residual Cl, the Fe fraction after isolation was evaporated to dryness at 95 °C and dissolved in 0.7 M HNO$_3$. This step was repeated for twice. The obtained recoveries with the in-house Fe isotope standard (CAG-Fe) were ≥93%. In all digestion and chromatographic separation steps, Teflon beakers with appropriate sizes and acid-clean polyethylene centrifuge tubes were used.

Fe has four stable isotopes ($^{54}$Fe, $^{56}$Fe, $^{57}$Fe, and $^{58}$Fe). The ratios of $^{57}$Fe/$^{54}$Fe and $^{56}$Fe/$^{54}$Fe were measured by using a Nu Plasma II MC-ICP-MS (Wrexham, UK) equipped with 16 Faraday cups. The instrument was operated at a medium mass resolution. All samples were diluted with HNO$_3$ solution (0.7 M) to obtain the intensities of $^{56}$Fe ~6 V. The optimized parameters of the instrument are given in Supplementary Table 5. The intensities of $^{56}$Fe obtained with the blank HNO$_3$ solution were <0.02 V, indicating that the background interference to the isotopic analysis were negligible. All samples were measured in at least two parallel experiments. The Fe isotopic composition of samples are reported using a commonly used $\delta$ value ($\delta^{56}$Fe in ‰) relative to the corresponding standard materials (IRMM-634) as follows:

$$\delta\,^{56}\mathrm{Fe} = \left( \frac{(^{56}\mathrm{Fe}/\,^{54}\mathrm{Fe})_{\text{sample}}}{(^{56}\mathrm{Fe}/\,^{54}\mathrm{Fe})_{\text{standard}}} - 1 \right) \times 1000‰ \qquad (1)$$

$$\delta\,^{57}\mathrm{Fe} = \left( \frac{(^{57}\mathrm{Fe}/\,^{54}\mathrm{Fe})_{\text{sample}}}{(^{57}\mathrm{Fe}/\,^{54}\mathrm{Fe})_{\text{standard}}} - 1 \right) \times 1000‰ \qquad (2)$$

$\Delta^{56}$Fe represents the difference in $\delta^{56}$Fe between two samples. The standard-sample-standard bracketing method was applied to correct the mass bias[43]. Meanwhile, in-house standards (CAG-Fe) were chosen as quality control samples. In this way, a $\delta^{56}$Fe value of 1.56 ± 0.1‰ (mean ± 2 s.d., $n = 14$) was obtained for the CAG-Fe solution, which showed no significant bias compared with the reported value in the literature[55]. These results demonstrated that our method for Fe isotopic analysis was highly accurate and precise.

**Reporting summary.** Further information on research design is available in the Nature Research Reporting Summary linked to this article.

## Data availability

The source data underlying Figs. 1a–f, 2d, 2g, 2u, and 4 and Supplementary Figs S1, S2, S3, S6, S9, S10, and S11 are provided as a Source Data file. Other data are provided in the Supplementary information or available from the corresponding author upon reasonable request.

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

## Acknowledgements

This work was financially supported by the National Natural Science Foundation of China (no. 21825403, 91843301, 21976194, 21904134, 91543105), the Chinese Academy of Sciences (no. QYZDB-SSW-DQC018, ZDBS-LY-DQC030), the Sanming Project of Medicine in Shenzhen (no. SZSM201811070), and the National Key R&D Program of China (2017YFC1309300). We thank Ms. Binbin Xiang from SIAT of CAS for help with sample collection.

## Author contributions

Q. Liu conceived and designed the research; G. Jiang supervised the project; Q. Luo, R. Chen, Y. Zhuansun, and J. Jiang collected the human serum and PE samples; D. Lu performed most of experiments; W. Wang, X. Yang, and L. Zhang helped with the particle characterization and MC-ICP-MS measurements; X. Liu helped with the toxicological assays; F. Li helped with the sample collection; Q. Liu and D. Lu analyzed the data; Q. Liu and D. Lu wrote the paper.

## Competing interests

The authors declare no competing interests.
