## [Peer Review File · Nature Communications]

Reviewers' Comments:

Reviewer #1:

Remarks to the Author:

Lu and colleagues isolate nanoparticles (NP) from human serum and pleural effusate (PE) samples and carry out physicochemical 'finger-printing' of the NPs to ascertain if these particles are exogenous (from inhalation to air pollution).

This is a very interesting study, with a very nice integration of complex microscopy and chemical analysis using human biological samples. The area is novel and important. While I confess that there is still some doubt in my mind as to whether the particles identified are from inhaled origin, a compelling case is made based on good scientific data.

I have a selection of suggestions that think should be addressed in the manuscript:

1- Line 51. The ability of nanoparticles to translocate is more than just a hypothesis: it has been comprehensive shown to occur (in animals at least) by the work of Wolfgang Kreyling. This body of work needs to be referenced and discussed. Recent translocation work by Miller et al (ACS Nano 2017; albeit using non-ambient gold nanoparticles) and Calderon-Garciduenas (Environ Pollut 2019) should also be mentioned. The latter in particular shows some similar methods to those in the current study to show that exogenous nanoparticles are found in the heart (presumably via the blood – thus, consider toning down statement on lines 55-57. Line 57 add 'ambient' before UFP).

2- Line 70 – It is stated that PE normally comes from the circulatory system. This needs to be expanded and made more implicit. Could the NPs from the PE come directly from the lung rather than the blood, especially if there is lung inflammation in response to NPs leading to increased permeability? This ultimately this does not change the origin-fate of the NPs, it will have implications for biological mechanisms and biological corona of NPs.

3- Lines 82-90: if possible please give some further indication as the likely the accuracy of the various techniques and potential losses. Given indications as to whether there are likely to be thresholds in the measurements for both particle number, mass and size estimates.

4- Arguably, it is possible that the NPs in the serum and PE could have been derived from ingested food and water (or even the skin) rather than from inhalation. I agree that the lung is the more likely route, but this possibility needs to be stated.

5- Methods – Given the nature of the work, it is a concern that these NPs are contaminants from the process of the methodology rather than inhaled NPs from biological samples. The authors state that care has been taken to avoid this, but the steps taken at each stage of the protocol needs to be stated.

6- Did the authors look for NPs in other organs and blood vessels (see line 101)? It should be emphasised in the Discussion that estimates of NPs represent a 'snap-shot' of particle movement around the body and that organ accumulation may be different for different particle types.

7- The authors note that many of these techniques could not be used for elements that occur naturally in the body. These elements (e.g. carbon, nitrogen, sulphate) will form a large proportion of the inhaled particle mass from sources of interest, and likely drive many toxic effects. That these particles are not addressed in the present study needs to be emphasised.

8- Toxicology assay. It should be highlighted that only a single assay with a single cell type has been used to address the potential for the recovered NPs to have health effects. Furthermore, the processes of isolating the particles is highly likely to change their toxicity. State if 'toxic effect' was linked to specific chemicals (line 111). Many other aspects will influence particle toxicology as well as chemical composition, e.g. size, charge, redox activity, shape, protein corona, cellular uptake, fate. Line 203 – tone down the phrase "significant toxic effects"

Minor comments

The authors should consider a slight modification of the title to include some reference to air pollution so it maximises the audience reached.

Line 46. Add cerebrovascular disease/stroke as a major contributor to the mortality associated with air pollution.

Line 66 – Further describe the environment where the volunteers are from, in particular, potential notable sources of pollution.

Line 88-90 – There is some doubt as to whether nanoparticles in the range of 30-100 nm are too big to translocate given what is known about natural barriers, pore sizes and mechanisms. This deserves some mention in the Discussion. Suppl Fig 6 – is a 400 nm particle really likely to translocate?

Throughout, make sure that where percentages are used it is clear if this is based on particle mass or particle number. The authors may wish to comment on the implications of these different metrics.

Many of the readers will be unfamiliar with the chemical analysis and microscopy techniques used in the study (indeed I am myself, so I cannot comment on limitations for these methods). The manuscript is clearly written, but where possible, take every opportunity to clearly state what each technique is measuring and why it is advantageous to use it in the current context.

Line 183 – state what 'residual PE' is, and if it can be ascertained that NPs in this sample are not due to incomplete separation.

Line 201 – specify carefully what the authors mean by combustion-derived? Would this be considered to include brake wear particles from friction?

Line 208 – It should be made clear that there is little doubt that UFPs are important to the health effects of air pollution. That UFPs are not included as a criteria pollutants is largely due to practical limitations of widely measuring in environment

Line 209-211 – I personally feel this sentence should be toned down.

Line 214 – This approach is very valuable, although I am not sure of the practicalities of its use in risk assessment.

Line 237 – How thoroughly was the participant's historical and current exposure to scrutinised? Were specific questions asked to ascertain this (participants may not be fully aware of potential sources of pollutants)?

Line 262 – Does the nanosight technique take into account particle agglomeration in suspensions?

A text description of the video file is needed.

Supplementary Figures – change *s on figures to symbols/numbers/letters to avoid confusion with degree of significance these symbols can represent.

Suppl Fig 3 – Did the two participant samples with higher levels of endotoxin produce greater effects in the 'toxicity assay'?

Reviewer #2:

Remarks to the Author:

Review of Lu et al. manuscript

Goal and Novelty:

The authors wished to find out as to whether inhaled ambient nano-sized particles will translocate into blood circulation in humans. This is an unsolved problem long discussed within the scientific community. For example, Nemmar et al. (2002) concluded from their studies in human subjects that inhaled Technegas (5-10 nm nanoparticles with 99mTc label) did translocate to the blood. Quite in contrast, Mills et al. (2006) did not confirm these findings in their Technegas inhalation study in human subjects, and pointed out a number of deficiencies in the Nemmar et al. study. A novel approach by Lu et al. presented in this paper is based on analyzing not only blood (serum) samples but also pleural effusates (PE) to find and characterize nanoparticles (NPs) in those fluids using novel high resolution EM/STEM imaging and EDS and EELS analysis. Their key assumption is that any NP in PE did originate in the blood circulation, confirming that ambient NPs in PE must have translocated into the blood circulation.

Strong Points:

Lu et al. used ultrahigh resolution imaging coupled with EDS and EELS analysis to carefully characterize nanoparticles detected in serum and pleural effusate of patients who live in a typical polluted region in China. This is the first study to apply this modern new technology to determine uptake of inhaled nanoparticles into the blood circulation of patients. Resulting images are impressive, although the interpretation of the results require careful reassessment as discussed below.

Weak Points, Suggestions for Revisions:

As evidence for their premise of the blood circulation origin of NPs in PE, the authors cite Song et al., 2009, and Andersen, 2005. The Song paper does not propose the circulation as source for the polyacrylate NPs they detected in PE, nor did Song et al. provide data that these NPs had been in the workplace air inhaled by the female subjects, who developed severe lung damage; quite to the contrary, Song et al. discussed the induced severe lung damage as reason for the distribution and appearance of the inhaled NPs in pulmonary epithelial and mesothelial cells. Gilbert (2009), cited by Lu et al. in this manuscript, pointed out problems in the Song paper, which, however, were not mentioned by Lu et al. Furthermore, the key reference (Andersen, 2005) cited by Lu et al. as evidence that "NPs in PE may be used as a proxy for those in blood" does not make sense and must be an error: This reference is a superficial review of a textbook, describing not the context, but mainly the physical characteristics of the book, such as weight, dimensions, number of pages, words/page, which cannot be taken seriously and has nothing to do with the topic of NP characterization.

The authors apparently lack detailed knowledge of particle inhalation physiology and toxicology which is not only evidenced by confusing statements of PM_{2.5} and ultrafine particles (UFP); but is more specifically obvious by the disregard of considering the normal bio-distribution of inhaled particles in the lung when interpreting results. As is well discussed by Donaldson et al. (2010), one clearance mechanism of inhaled fibrous and non-fibrous particles deposited in the lung is via lymphatic pathways to the pleural cavity and to pulmonary and mediastinal lymph nodes. Yes, Lu et al. are correct to conclude that PE NPs are reflecting inhaled airborne NPs, but not because they originate from the blood circulation, but come directly from the deposits in the pulmonary alveolar region. This is consistent with the finding (lines 94, 95) of much higher NP concentration in PE vs serum reported by the authors.

The patient subjects are divided into healthy and diseased. It is rather confusing though to see in Supplementary Tables 1 and 2 that all patients were diseased, contradicting the statement (lines

235, 236) that "the healthy individuals presented no clinical evidence of diseases." What else than clinical tests did reveal the different kinds of disease in each patient? There are 20 patients with different types of cancers, COPD, pneumonia, TB, heart failure, POEMS, all patients were diseased according to Table 1. Who are the healthy patients?

The authors took appropriate precaution to avoid any contamination of serum and PE samples when they were handled in an ultra-clean laboratory environment. However, nothing is said how potential contamination was avoided during the blood and PE sample collection from the patients. Given that the presence of NPs in PE does not indicate a circulatory origin (see comment above), it leaves only the serum samples as direct indicator for confirming NP translocation, so ultra-clean blood collection is essential.

Confirming the concordance of NP characteristics in serum and PE by structural finger-prints appeared to be difficult due to the much lower NP concentration in serum. To establish, therefore, that structural fingerprint findings of Fe and O in serum indicate environmental origin requires stronger evidence. Crystallinity may not separate biogenic from abiogenic Fe-oxides.

It has to be considered that under inflammatory conditions in the lung (e.g., pneumonia, COPD) the epithelial integrity (tight junctions) is compromised and transfer of solutes and NPs occurs in both directions. Biotransformation processes of NPs in the lung and tissues are also to be expected. Can singular findings in a PE sample of one patient (P12) and a serum sample of one other patient (P32) (lines 170/171) be generalized as evidence That these are counterparts in serum and PE?

The assumption expressed in this manuscript that results of the authors' in vitro RAW studies show the true pro-inflammatory activity of the particles in the human body is an overstatement. The exposure concentration of the cell medium – used as metric - is not equivalent to the actual dose to a cell; also, using one concentration only will not allow to establish dose-effect relationships.

References:

Donaldson, K., F. A. Murphy, R. Duffin and C. A. Poland (2010). "Asbestos, carbon nanotubes and the pleural mesothelium: a review and the hypothesis regarding the role of long fibre retention in the parietal pleura, inflammation and mesothelioma." *Part Fibre Toxicol* 7(1): 5.

Mills, N. L., A. N., S. D. Robinson, A. Anand, J. Davies, D. Patel, J. M. de la Fuente, F. R. Cassee, N. A. Boon, W. MacNee, A. M. Millar, K. Donaldson and D. E. Newby (2006). "Do inhaled carbon nanoparticles translocate directly into the circulation in humans?" *Am.J. Respir. Crit. Care Med.* 173: 426-431.

Nemmar, A., P. H. M. Hoet, B. Vanquickenborne, D. Dinsdale, M. Thomeer, M. F. Hoylaerts, H. Vanbilloen, L. Mortelmans and B. Nemery (2002). "Passage of inhaled particles into the blood circulation in humans." *Circulation* 105: 411-414.

Reviewer #3:

Remarks to the Author:

General Comments

The authors have identified an important aspect of pollution uptake into systemic circulation derived from ultra-fine particulate matter that is not typically addressed with air pollution standards of PM_{2.5}.

The critical assessment of the ultra-fine particulate NPs involving chemical, structural and size-related characteristics has been addressed and shown for serum and pleural effusion.

Major Claims:

Linking the sources of ultra-fine particulates such as combustion-derived or traffic related emission NPs to the types of NPs found in serum and PE was succinctly performed and is comparing the liquid-containing nanoparticle host "serum/PE" versus previously published tissue-containing hosts (brain, liver, spleen etc.). Identifying NP in tissue and comparing with exogenous source materials has been widely published and the current work focuses on systemic NPs and their origins. Nanoparticle uptake into blood has been studied previously, but PE as a host of NP is innovative and was detailed in the manuscript.

Understanding the role of NP in systemic circulation and their origins will be of importance to nanotoxicology and risk assessment in particular and environmental pollution-derived disease developments and public health in general.

Novelty Aspects:

The authors addressed various sources like coal combustion fly ash and automotive exhaust catalyst particles as exogenous matter that translocated to PE via serum uptake using particle shapes, isotopic signatures of Fe and metal concentrations in PE. One important area that was overlooked or not addressed is the dynamic nature of NPs and if this will also be the case for serum and PE. It should be considered whether the translocated particles will be subject to bio-transformations including partial dissolution, reformation to secondary particles (in vivo formation). This should at least be pointed out in the discussion part.

Study conclusions: The overall conclusions are well in line with previous studies that investigated the in vivo translocation and toxicity aspects of nanoparticles after exposure. However, the uptake of NPs into PE from systemic circulation has a significant novelty aspect because it relates to nanoparticles in liquid medium instead of tissue interactions and uptake. The authors demonstrate using HAADF STEM and EELS that the chemical and structural nature of the NPs can be linked to exogenous matter. The Fe isotope analyses of PE containing nanoparticles and serum is also novel and allows to pinpoint to the origins of the ultra-fine particulate matter and this aspect is original, innovative and deserves publication.

Review Summary: The manuscript is original but requires several modifications and they are itemized in detail under "Specific Comments" below:

Specific Comments:

Line 38:only in recent decades did it become one of the leading global health risks due to dramatically increased anthropogenic sources. Need Reference(s)

Line 42:More than 90% of the global population are living in polluted air. Need reference for population data

Line 44: ...Long-term exposure to PM2.5 is thought to increase mortality and morbidity and shorten life expectancy by causing cardiovascular and respiratory diseases, such as respiratory infections, chronic obstructive pulmonary (COPD), heart attack, and lung cancer. Reference is missing.

Line 66: The NPs in serum can directly reflect the internal exposure to UFPs. This statement needs to be reworded since it suggests that a direct observation of NP level in serum and dose exposure effects are already known or accepted.

Line 72: ...The extracted NPs were purified by enzymatic hydrolysis Need to clarify that the NPs were extracted from PE.

Line 74:multi-fingerprints (including elemental, structural, and natural isotopic fingerprints) Not sure why the authors refer to "natural isotopic" - if all NPs trapped in PE are characterized, then

there is no distinction needed here for "natural".

Line 75:It should be noted that PE samples can only be clinically collected from patients with some specific diseases. So, in this study, both patients and healthy subjects were recruited, and the serum samples were collected from both healthy subjects and patients to compare the NP levels in the human body between healthy subjects and patients. This is contradicting the earlier explanations that PE samples would be used to extract NPs. If PE samples are only taken from patients and not for healthy subjects then there is only a comparison possible for serum. Specify

Line 88:... Concentration and size of NPs in human serum. As shown in Fig. 1a, nanoparticle tracking analysis shows that NPs are ubiquitous in all of the human serum samples with the concentration ranging from 1.4×10^8 to 1.0×10^{10} particles mL⁻¹ (mean value 2.6×10^9 particles mL⁻¹). The size distribution shows large polydispersity and individual differences within a few to hundreds of nanometers. Such a particle size range actually exceeds the commonly supposed threshold for atmospheric particulate matter to penetrate through the pulmonary alveoli ($< 0.1 \mu\text{m}$), suggesting that the penetrability of particulate matter may be stronger than previously thought.

- 1) The statement that NPs are ubiquitous in all of the human serum samples suggests that NPs were classified as coming from different sources. This needs to be restated since the authors do not have this information.
- 2) Concentration ranges for NPs are strongly dependent on the potential agglomeration tendencies of the NP in serum. Furthermore, the separation of NP will also affect dispersion, agglomeration which can change the concentration ranges for particles in serum.
- 3) The size distribution of NP in serum depends on agglomeration tendencies since agglomerated particles will be measured as a "larger particle". This needs to be discussed here when talking about "polydispersity" and size differences.
- 4) If particles agglomerate and deagglomerate in serum (and during the extraction procedures) then the statement that penetrability of particulate matter may be stronger than previously thought is not proven. The complexity of particle agglomeration tendency needs to be considered here. Also, the authors did not consider that NP can partially dissolve which can reduce size and other factors.

Line 98: ...(ii) the NPs in PE are less polydisperse than those in serum (nearly all of the NPs in PE are constrained within 50-200 nm), suggesting that the pleura may have a sieving effect for NPs due probably to the deposition of some large particles at the vascular walls¹². Again, this is an unproven statement by the authors: Deagglomeration potential has not been addressed; Partial Solubility of NP particles as a result of residence time in PE has not been addressed; NP transformations in PE have not been addressed as a potential factor to affect a smaller particle size range in PE. Also, there would be an "accumulation effect" where NP are either stored in PE or removed from PE.

Line 105:.... Despite that, we find that the PE-derived NPs can cause significant pro-inflammatory effects at the real doses as in the human body after excluding the effect of endotoxin for all PE samples (Fig. 1e and Supplementary Fig. 3), suggesting that the NPs in the human body have significant health risks. This paragraph is completely confusing. It is not clear how the authors obtain the information on pro-inflammatory effects at "real doses"? Need to elaborate. In introduction it was stated that extracted NPs were digested to eliminate protein coatings on the particle surfaces etc which affects the NP reactivity and their in vivo toxicity. To suggest that NPs have significant health risks is generally known and documented, but the actual pro-inflammatory effects of NPs in PE depend on a series of factors which are not addressed by the authors.

Line 111:.... Thus, the toxic effect of the NPs may be more relevant to their chemical nature. The chemical nature of the NP in PE and serum are very divers and to suggest that the toxic effects of the NPs may be more relevant to their chemical nature is far too vague.

- 1) Surface area effects of NPs are not addressed;
- 2) Redox Potential of NPs is not addressed
- 3) Solubility effects are not addressed

Line 113:.... chemical multi-fingerprints: What exactly do the authors mean by "multi-fingerprints"?

Extreme diversity in chemical fingerprints?

Line 114:.... Besides the high-abundance elements in organisms. Are the authors suggesting that NPs in PE and serum are composed of elements that are constituents of "high-abundance elements in organisms"? The analyses should focus on the actual NPs extracted from PE and serum. In Figure 1f the elemental concentrations of Fe >> Mn > Ti > Ba > Sr would reflect elements that are common constituents. The question that needs to be addressed is: What kind of NPs are high in Fe (oxides, phosphates, oxyhydroxides; sulfates etc..) and are the Fe particles magnetic (Fe₃O₄)?and what kind of NPs contain Mn, Ba, Sr, Ti etc. Are these NPs oxides or some other compounds?

Line 117: ... Note that Al, Fe, Ti, Mn, Ba, and Sr, which are rock-forming elements or 118 relatively abundant elements in the Earth's crust, show relatively high concentrations in the 119 NPs (Fig. 1f). Such elemental fingerprints suggest that the NPs are of external abiogenic 120 sources. This is not correct! Fe (iron) can form biomineralized ferritin NPs from endogenous iron source and the presence of Fe in serum and PE could be derived from either endogenous or exogenous or both sources. This needs to be clarified in the text.

Line 120:.....Remarkably, Pt, a characteristic element in vehicle exhaust catalysts, is found in some samples from benign patients (with no application of Pt-containing agents), suggesting that some NPs may originate from vehicle exhausts¹³. Pt is also used in many other catalyst applications besides automotive. I would recommend not to make extrapolations in the Results section.

Line 124: The elemental fingerprints of NPs also show large individual differences (Fig. 1g), which can partly explain the toxic effect of the NPs. I think this sentence needs to be reworded. Figures 1 e and g together indicate how NPs for different samples (P) show elemental tendencies that are linked to an inflammatory response: For example P34 and P36 in Figure 1g have high Vanadium which is indicated in Figure 1e to correspond to an elevated inflammatory response.

Line 134:... Abundant Fe-bearing NPs are found, as Fe makes up 0.17‰-12.27‰ of the mass of the NPs. Need to refer to the structures of the Fe-NPs as observed in HAADF STEM.

1) A number of spherical NPs consisting of Fe and O are found (section I in Fig. 2a; Fig. 2b-d). Indexing of the lattice fringes and the FFT pattern of the particle are consistent with the magnetite crystal structure (Fig. 2e-f).

Line 129: Structural fingerprints of NPs in PE. The structural information is gained from the HAADF-STEM (Morphological; Crystallographic) . The EDS and EELS data is chemical and electrochemical information and the heading "Structural Fingerprints" needs to be expanded.

Line 140: Recall that endogenous magnetite particles formed via in situ crystallization within the 8-nm-diameter core of ferritin should have a euhedral angular crystal morphology^{4,14,15}. Biomineralized Ferritin NPs that form in the ferritin nanocage are actually not angular and are composed predominantly of iron oxyhydroxide nanoparticles, This needs to be corrected. Only ferritins that form in vivo should be considered and not iron oxide nanoparticles that are synthesized using a ferritin precursor nanocage.

Line 142:... Thus, the magnetite NPs observed here contrast with the biogenic ones but resemble the pollution NPs found in the human brain reported previously⁴. Several of the brain -containing magnetite from reference 4 also have spherical and angular morphologies. The main difference between the ferritin derived iron oxide NPs and those found in the brain study were the larger particle sizes of the magnetite! It would be good if the authors refer to the particle size of the magnetite found in the PE and serum samples. The example in Figure 2 shows spherical magnetite NPs.

Line 146:....Besides magnetite, we also observe other Fe-bearing NPs such as rounded crystal NPs consisting of Fe, Mn, and Ni (Fig. 2h-j and Supplementary Fig. 5) that resemble Fe-Mn-Ni alloy particles existing in PM_{2.5} emitted from ferroalloy plants. Are these oxides or metal spheres? This is not clear in the text or in Figure 2. Fe, Mn, Ni are very common in fly ash from coal combustion plants. They typically occur as "fly ash spheres" with a high percentage of transition metal oxides that are contained in a glassy (Si, Al) matrix. To be derived from a ferroalloy plant the NP would be reduced metals or metallic. If the authors relate the Fe-Mn-Ni NPs to ferroalloy plants they need to distinguish them from fly ash spheres.

Line 151:.... Elemental Cu is scarcely present in natural environment or human body, and it is most likely to be emitted by electric motors in indoor environments. Cu-solution is typically used as a wood preservative and has been shown to form nanoparticles inside the lumber. Any lumber cutting or manufacturing process could set free NPs and the authors need to consider this source as well.

Line 155: We also find hexagonal Ti-bearing particles (Fig. 2p). Lattice indexing and EDXS mapping indicate that they are crystal TiO₂ NPs (Fig. 2q-s), which are one of the most produced engineered nanomaterials. Should also mention that industrially produced nano-TiO₂ often is spherical in nature or occurs as nanorods which have a hexagonal crystal lattice.

Line 169:... In addition, we identified a type of special fusiform NPs with a mineral-like elemental composition (C, O, Ca, and P). Not sure what fusiform NPs means. May need to elaborate. The composition C, O, Ca and P may be suggestive of a) Calcium Carbonate and Calcium Phosphate which can have both exogenic or endogenic origins.

Line 206:... The chemical multi-fingerprinting shows that the NPs mainly originate from external particulate pollution, particularly, combustion-related particulate emission. I would suggest to include "structural" fingerprinting since fly ash spheres can be identified and this would suggest some combustion sources.

Line 217:.... which also needs to take into account the effect of protein corona on the NPs in the future studies And also needs to take into account solubility of NPs from combustion sources after uptake into serum and PE.

Line 218: ... The occurrence states of exogenous NPs in more human organs need to be studied to fully understand the dynamics and life cycle of ambient UFPs in the human body; Also, the authors should mention that NPs in PE and serum will have a residence time which will affect particle transformations and solubility and clearance.

Response to Reviewer Comments

We really appreciate the great efforts that reviewers have made to improve the quality of our manuscript entitled “Chemical multi-fingerprinting of exogenous nanoparticles in human serum and pleural effusion” (NCOMMS-19-21077). We have now finished the revision according to reviewers’ comments and made substantial revisions to the manuscript. Each point raised by the reviewers has been carefully considered and fully addressed beneath. The changes in the manuscript are marked using red font. In this response letter, the reviewers’ comments copied verbatim beneath are in italic blue font and the author responses are in black font, and page numbers refer to those in the revised manuscript.

Response to Reviewer #1:

Lu and colleagues isolate nanoparticles (NP) from human serum and pleural effusate (PE) samples and carry out physicochemical ‘finger-printing’ of the NPs to ascertain if these particles are exogenous (from inhalation to air pollution). This is a very interesting study, with a very nice integration of complex microscopy and chemical analysis using human biological samples. The area is novel and important. While I confess that there is still some doubt in my mind as to whether the particles identified are from inhaled origin, a compelling case is made based on good scientific data. I have a selection of suggestions that think should be addressed in the manuscript:

Question: *1-Line 51. The ability of nanoparticles to translocate is more than just a hypothesis: it has been comprehensive shown to occur (in animals at least) by the work of Wolfgang Kreyling. This body of work needs to be referenced and discussed. Recent translocation work by Miller et al (ACS Nano 2017; albeit using non-ambient gold nanoparticles) and Calderon-Garciduenas (Environ Pollut 2019) should also be mentioned. The latter in particular shows some similar methods to those in the current study to show that exogenous nanoparticles are found in the heart (presumably via the blood – thus, consider toning down statement on lines 55-57. Line 57 add ‘ambient’ before UFP).*

Answer: Thanks very much for your suggestion. We have corrected the related statements (e.g., page 3 line 47, “hypothesis” to “point”; line 49, “hypothesized” to “thought”) and cited the relevant references in the revised manuscript. The Kreyling works have been introduced and cited in the Introduction (please see page 3 line 55 and Ref. 8 and 10-13). Recent works by Miller et al. and Calderon-Garciduenas et al. have also been discussed (please see page 4 line 57-59 and Ref. 15-16). Furthermore, the statement you mentioned in line 55-57 has been toned down or removed to be more accurate.

Question: 2- Line 70 – *It is stated that PE normally comes from the circulatory system. This needs to be expanded and made more implicit. Could the NPs from the PE come directly from the lung rather than the blood, especially if there is lung inflammation in response to NPs leading to increased permeability? This ultimately this does not change the origin-fate of the NPs, it will have implications for biological mechanisms and biological corona of NPs.*

Answer: Thanks very much for your valuable comments. According to your comments, we have carefully re-considered the influx of NPs in the PE and re-interpreted our results in the revised manuscript. We quite agree that, although PE normally comes from the circulatory system, the NPs in PE may also directly come from the lung, especially under the inflammatory conditions (*Part. Fibre Toxicol.* 2010, 7, 5; *Lung* 2001, 179, 397-413). Therefore, we have added the possibility of the direct translocation of UFPs from the lung to the PE in the revised manuscript. This means that the NPs in the PE may come both directly from the lung and the blood circulation. This point actually enhances the major claim of this study that the UFPs in the PE and blood circulation are of same origin from ambient UFPs and thus the PE can be used as a host to study the inhaled UFPs. Please see page 4 line 72-79. This point is also used to better interpret the experimental results. Please see page 6 line 109-111; page 10 line 193-194.

Question: 3- *Lines 82-90: if possible please give some further indication as to the likely the accuracy of the various techniques and potential losses. Given indications as to whether there are likely to be thresholds in the measurements for both particle number, mass and size estimates.*

Answer: Thanks for your suggestion. In this revision, we have added more description for each technique used in this study and their possible losses (e.g., components of carbon, nitrogen, and sulphate of inhaled UFPs). Please see page 5 line 90-91 and 99-105; page 7 line 138-141; and page 12 line 251-256. Specifically, for particle number and size measurement by NTA, the working ranges for particle concentration and size are 10^7 - 10^9 particles mL^{-1} (the sample can be diluted or enriched before measurement if the particle concentration is beyond this range) and 10-2000 nm, respectively, which has been clearly specified in the revised manuscript. Please see page 5 line 99-105 and page 16 line 308-311.

Question: 4- *Arguably, it is possible that the NPs in the serum and PE could have been derived from ingested food and water (or even the skin) rather than from inhalation. I agree that the lung is the more likely route, but this possibility needs to be stated.*

Answer: Thanks for your suggestion. We agree that the NPs still keep the possibility of deriving from ingested food or water (or even the skin), and this possibility has been mentioned in the Discussion. Please see page 11 line 227-229.

Question: 5- *Methods – Given the nature of the work, it is a concern that these NPs are*

contaminants from the process of the methodology rather than inhaled NPs from biological samples. The authors state that care has been taken to avoid this, but the steps taken at each stage of the protocol needs to be stated.

Answer: Thanks for your suggestion. We are fully aware that it is extremely important to exclude any possible contamination in this study. So, we have taken every opportunity to eliminate any possible contamination sources during the whole sampling and experimental procedures, e.g.,

- i) All sample collection was conducted in biological laminar air flow wards using ultra-clean disposable devices, and after collection the samples were immediately tightly sealed until analysis;
- ii) All devices and containers used in the experiments were thoroughly washed with particle-free sterile water for several times before use;
- iii) More importantly, control experiments with particle-free solvents only was conducted to ensure no contamination during the sample preparation procedures (particle number below detection limit in NTA and no pollution particulates were found under electron microscopes).

In this way, we believe that the possible contaminants could be fully eliminated. All the measures taken at each step are clearly stated in the Methods. Please see page 14 line 269-272 and 280-282; page 15 line 297-300.

Question: *6- Did the authors look for NPs in other organs and blood vessels (see line 101)? Is should be emphasised in the Discussion that estimates of NPs represent a 'snap-shot' of particle movement around the body and that organ accumulation may be different for different particle types.*

Answer: Thanks for your suggestion. In the present study, we focused on the PE and serum and did not investigate other organs or blood vessels. It indeed only represents a “snap-shot” of particle movement in the human body and more organs will be considered in our future studies to fully understand the dynamics and lifecycle of UFPs in the human body. This point has been emphasized in the Discussion in the revised manuscript. Please see page 12 line 246-251.

Question: *7- The authors note that many of these techniques could not be used for elements that occur naturally in the body. These elements (e.g. carbon, nitrogen, sulphate) will form a large proportion of the inhaled particle mass from sources of interest, and likely drive many toxic effects. That these particles are not addressed in the present study needs to be emphasised.*

Answer: Thanks for your suggestion. We have emphasized in the Discussion the absence of some elements (e.g., carbon, nitrogen, sulphate) in the present study due to the limitations of

the techniques used and their specific requirements for samples. Please see page 13 line 254-259.

Question: 8- *Toxicology assay. It should be highlighted that only a single assay with a single cell type has been used to address the potential for the recovered NPs to have health effects. Furthermore, the processes of isolating the particles is highly likely to change their toxicity. State if 'toxic effect' was linked to specific chemicals (line 111). Many other aspects will influence particle toxicology as well as chemical composition, e.g. size, charge, redox activity, shape, protein corona, cellular uptake, fate. Line 203 – tone down the phrase “significant toxic effects”.*

Answer: Thanks very much for your comment. After careful consideration and taking into account the reviewers' comments, we have removed the toxicological assay from the revised manuscript based on the following reasons:

- i) The toxicological results obtained here are very rough with only a single cell type in a single assay, which is difficult to reflect the real doses and toxic effects of particles in the human body;
- ii) Due to the NP extraction process used in the present study, the surface properties (e.g., surface chemistry, redox potential, and solubility) of NPs might have been altered compared with those in the human body, which may greatly affect the toxicity of the NPs;
- iii) Moreover, the main objective of this study is to probe and characterize exogenous UFPs in the human serum and PE, and, therefore, the toxicological effect of the NPs is likely beyond the scope of the present study. It should deserve a more comprehensive study and to be published as a separate paper.

Therefore, to make the present paper more reasonable, we think that it should be better to be more concentrated on the characterization of NPs in the PE and serum. Anyway, if you think that the toxicological assay should be kept in this paper, we will be happy to add it into the paper again with proper revisions.

Minor comments:

Question: *The authors should consider a slight modification of the title to include some reference to air pollution so it maximises the audience reached.*

Answer: Thanks for your suggestion. We have changed the “nanoparticles” to “ultrafine particles” which is more frequently used in aerosol toxicology in the title.

Question: *Line 46. Add cerebrovascular disease/stroke as a major contributor to the mortality associated with air pollution.*

Answer: In page 3 line 46, “stroke” has been added to the text.

Question: *Line 66 – Further describe the environment where the volunteers are from, in particular, potential notable sources of pollution.*

Answer: The notable sources of air pollution in the region where the volunteers were recruited from (i.e., the Pearl River Delta region of China) have been described in the revised manuscript. Please see page 4 line 67-69.

Question: *Line 88-90 – There is some doubt as to whether nanoparticles in the range of 30-100 nm are too big to translocate given what is known about natural barriers, pore sizes and mechanisms. This deserves some mention in the Discussion. Suppl Fig 6 – is a 400 nm particle really likely to translocate?*

Answer: Thanks for your comment. In this revision, we have moved the discussion about the threshold of particle size for translocation to the Discussion section, and, according to your suggestion, mentioned the doubt about the translocation of particles of 30-100 nm. Please see page 12 line 232-236.

In this study, we have observed a number of particles with sizes in the range of 30-100 nm or even $> 0.1 \mu\text{m}$. On one hand, particles with sizes in such ranges have also been found in other reports, e.g., in human heart by Calderon-Garciduenas et al. (*Environ. Res.* 2019, *176*, 108567) and in human brain by Maher et al. (*PNAS* 2016, *113*, 10797-10801). The translocation of large particles (e.g., 240 nm) via inhalation pathway has also been reported to occur in animals (*Cell Tissue Res.* 2003, *311*, 47-51). On the other hand, the potential agglomeration tendency of NPs in body fluids and in extraction process may also cause large variation in particle size. Therefore, we agree that further studies are still needed to determine whether large particles with sizes of hundreds of nanometer really likely to translocate via inhalation. Please see page 12 line 236-242 and Supplementary Fig. 5.

Question: *Throughout, make sure that where percentages are used it is clear if this is based on particle mass or particle number. The authors may wish to comment on the implications of these different metrics.*

Answer: Thanks for your suggestion. Throughout the manuscript, where the percentages are used their meanings are clearly indicated (in mass or particle number). Please see page 7 line 144.

Question: *Many of the readers will be unfamiliar with the chemical analysis and microscopy techniques used in the study (indeed I am myself, so I cannot comment on limitations for these methods). The manuscript is clearly written, but where possible, take every opportunity to clearly state what each technique is measuring and why it is advantageous to use it in the current context.*

Answer: Thanks for your kind suggestion. We have added more description about the techniques used in this study, including their capabilities and advantages. Please see page 5

line 90-91 and 99-100 for NTA; page 7 line 138-141 for HAADF-STEM with EDXS and EELS; and page 10 line 197-198 for isotopic analysis using MC-ICP-MS.

Question: *Line 183 – state what ‘residual PE’ is, and if it can be ascertained that NPs in this sample are not due to incomplete separation.*

Answer: The “residual PE” means the PE sample after particles were fully separated by repeated high-speed centrifugation at 10000 rpm, which has been specified more clearly in the revised manuscript. Please see page 10 line 198-199 and 202. The separation efficiency of NPs from PE has also been ascertained by recovery test with a silica nanoparticle standard reference material and NTA (recovery > 97%).

Question: *Line 201 – specify carefully what the authors mean by combustion-derived? Would this be considered to include brake wear particles from friction?*

Answer: The definition of “combustion-derived particles” is given in page 11 line 210-211. It means anthropogenic fine particles emitted through combustion processes, including vehicle exhausts, municipal solid waste incineration, industrial processes, and frictional heating of brake pads. Considering that frictional heating can also produce high-temperature, the brake wear particles from friction should also be included.

Question: *Line 208 – It should be made clear that there is little doubt that UFPs are important to the health effects of air pollution. That UFPS are not included as a criteria pollutants is largely due to practical limitations of widely measuring in environment.*

Answer: Thanks for your suggestion. The related statement has been revised to “UFPs have not been classified as a criteria pollutant in the National Ambient Air Quality Standard mainly due to practical limitations of current environmental measurement”. Please see page 11 line 229-231.

Question: *Line 209-211 – I personally feel this sentence should be toned down.*

Answer: This sentence has been revised to “This preliminary study provides a clear evidence in human for the systemic health effects of ambient UFPs”. Please see page 11 line 231-232.

Question: *Line 214 – This approach is very valuable, although I am not sure of the practicalities of its use in risk assessment.*

Answer: Thanks very much for your positive comment. We agree that more studies are needed to further demonstrate the practicalities of this technique in nanomaterial risk assessment, which are in progress in our lab.

Question: *Line 237 – How thoroughly was the participant’s historical and current exposure to scrutinised? Were specific questions asked to ascertain this (participants may not be fully aware of potential sources of pollutants)?*

Answer: Thanks for your question. We have indeed included specific questions in the survey about the historical and current careers of the participants to exclude any occupational exposure to hazardous materials.

Question: *Line 262 – Does the nanosight technique take into account particle agglomeration in suspensions?*

Answer: Thanks for your question. In NTA, to prevent potential particle agglomeration, a 0.05% (v/v) TWEEN-20 as a stabilizer was added to sample solution at a ratio of 1:25 (below the critical micelle concentration of TWEEN-20) to maintain the stability of NPs as previously reported (*ACS Nano* 2014, 8, 2439-2455). This has been clarified in the revised manuscript. Please see page 16 line 309-311.

Question: *A text description of the video file is needed.*

Answer: A caption for the Supplementary Video has been provided in the SI.

Question: *Supplementary Figures – change *s on figures to symbols/numbers/letters to avoid confusion with degree of significance these symbols can represent.*

Answer: According to your suggestion, the asterisks in Supplementary Fig. 1 and 2 have been changed to numbers to avoid confusion.

Question: *Suppl Fig 3 – Did the two participant samples with higher levels of endotoxin produce greater effects in the ‘toxicity assay’?*

Answer: As responded above, the toxicological contents including the original Supplementary Fig. 3 have been removed from the revised manuscript.

Response to Reviewer #2:

Goal and Novelty:

The authors wished to find out as to whether inhaled ambient nano-sized particles will translocate into blood circulation in humans. This is an unsolved problem long discussed within the scientific community. For example, Nemmar et al. (2002) concluded from their studies in human subjects that inhaled Technegas (5-10 nm nanoparticles with 99mTc label) did translocate to the blood. Quite in contrast, Mills et al. (2006) did not confirm these findings in their Technegas inhalation study in human subjects, and pointed out a number of deficiencies in the Nemmar et al. study.

A novel approach by Lu et al. presented in this paper is based on analyzing not only blood (serum) samples but also pleural effusates (PE) to find and characterize nanoparticles (NPs) in those fluids using novel high resolution EM/STEM imaging and EDS and EELS analysis. Their key assumption is that any NP in PE did originate in the blood circulation, confirming that ambient NPs in PE must have translocated into the blood circulation.

Strong Points:

Lu et al. used ultrahigh resolution imaging coupled with EDS and EELS analysis to carefully characterize nanoparticles detected in serum and pleural effusate of patients who live in a typical polluted region in China. This is the first study to apply this modern new technology to determine uptake of inhaled nanoparticles into the blood circulation of patients. Resulting images are impressive, although the interpretation of the results require careful reassessment as discussed below.

Weak Points, Suggestions for Revisions:

Question: *As evidence for their premise of the blood circulation origin of NPs in PE, the authors cite Song et al., 2009, and Andersen, 2005. The Song paper does not propose the circulation as source for the polyacrylate NPs they detected in PE, nor did Song et al. provide data that these NPs had been in the workplace air inhaled by the female subjects, who developed severe lung damage; quite to the contrary, Song et al. discussed the induced severe lung damage as reason for the distribution and appearance of the inhaled NPs in pulmonary epithelial and mesothelial cells. Gilbert (2009), cited by Lu et al. in this manuscript, pointed out problems in the Song paper, which, however, were not mentioned by Lu et al. Furthermore, the key reference (Andersen, 2005) cited by Lu et al. as evidence that “NPs in PE may be used as a proxy for those in blood” does not make sense and must be an error: This reference is a superficial review of a textbook, describing not the context, but mainly the physical characteristics of the book, such as weight, dimensions, number of pages, words/page, which cannot be taken seriously and has nothing to do with the topic of NP characterization.*

Answer: Thanks very much for your kind suggestions. Firstly, according to your suggestion, we have carefully re-considered the influx of the NPs in the PE and re-interpreted our results in the revised manuscript. The premature claim about the blood circulation origin of NPs in PE has also been corrected. Please see the detailed response to your next question.

Secondly, we have corrected the confusing references cited in the manuscript. The Song et al. paper and Gilbert paper are moved to the Results section and are properly introduced to support the direct translocation of UFPs to the PE. Please see page 6 line 111-113. The wrong reference of Andersen paper has been deleted from the revised manuscript, and, instead, as key references, we cite two papers describing the clearance mechanisms of particles from the lung (*Part. Fibre Toxicol.* 2010, 7, 5; *Lung* 2001, 179, 397-413). Please see Ref. 21-22 and page 4 line 72-75.

Question: *The authors apparently lack detailed knowledge of particle inhalation physiology and toxicology which is not only evidenced by confusing statements of PM_{2.5} and ultrafine particles (UFP); but is more specifically obvious by the disregard of considering the normal bio-distribution of inhaled particles in the lung when interpreting results. As is well discussed*

by Donaldson et al. (2010), one clearance mechanism of inhaled fibrous and non-fibrous particles deposited in the lung is via lymphatic pathways to the pleural cavity and to pulmonary and mediastinal lymph nodes. Yes, Lu et al. are correct to conclude that PE NPs are reflecting inhaled airborne NPs, but not because they originate from the blood circulation, but come directly from the deposits in the pulmonary alveolar region. This is consistent with the finding (lines 94, 95) of much higher NP concentration in PE vs serum reported by the authors.

Answer: We really appreciate your kind and valuable comments. Firstly, we have revised the definition of PM_{2.5} and UFPs in the Introduction to be more accurate. Please see page 3 line 39 and 50.

Secondly, according to your comments, we have carefully re-considered the influx of the NPs in the PE and re-interpreted our results in the revised manuscript. We quite agree that the NPs in PE may also directly come from the lung, especially under the inflammatory conditions, in addition to the exchange of particles with blood circulation. Therefore, in this revision,

- 1) We have clearly discussed the possibility of the direct translocation of UFPs from the lung to the pleural cavity. The key assumption has been narrowed to that the UFPs in the PE and blood circulation are of same origin from ambient UFPs and thus the PE can be used as a host to reflect the inhaled UFPs in the human body. This point is well supported by the fact that the NPs in the PE can come both directly from the lung and the blood circulation. Please see page 4 line 72-79.
- 2) We have removed the premature claim that the NPs in the PE are translocated via the blood circulation from the revised manuscript. The direct translocation of UFPs from the lung to the PE is used to better interpret the experimental results. Please see page 6 line 109-111; page 10 line 193-194.

Question: *The patient subjects are divided into healthy and diseased. It is rather confusing though to see in Supplementary Tables 1 and 2 that all patients were diseased, contradicting the statement (lines 235, 236) that “the healthy individuals presented no clinical evidence of diseases.” What else than clinical tests did reveal the different kinds of disease in each patient? There are 20 patients with different types of cancers, COPD, pneumonia, TB, heart failure, POEMS, all patients were diseased according to Table 1. Who are the healthy patients?*

Answer: We are sorry for the confusing description of the participants in the previous version where only patient information is provided in the Supplementary Table 1-2. In this revision, the information of all study participants including the healthy subjects (in Supplementary Table 2) has been provided in the SI. Please see Supplementary Table 1-3 in the SI.

Question: *The authors took appropriate precaution to avoid any contamination of serum and PE samples when they were handled in an ultra-clean laboratory environment. However, nothing is said how potential contamination was avoided during the blood and PE sample collection from the patients. Given that the presence of NPs in PE does not indicate a circulatory origin (see comment above), it leaves only the serum samples as direct indicator for confirming NP translocation, so ultra-clean blood collection is essential.*

Answer: Thanks for your suggestion. We are fully aware that it is essential to exclude any possible contamination in this study, so we have taken particular precautions to eliminate any possible contamination sources during the whole sampling and experimental procedures, e.g.,

- i) All sample collection was conducted in biological laminar air flow wards using ultra-clean disposable devices, and after collection the samples were immediately tightly sealed until analysis;
- ii) All devices and containers used in the experiments were thoroughly washed with particle-free sterile water for several times before use;
- iii) Control experiments with particle-free solvents only was conducted to ensure no contamination during the sample preparation procedures (particle number below detection limit in NTA and no pollution particulates were found under electron microscopes).

In this way, we believe that the possible contaminants could be fully eliminated. All the measures taken at each step are clearly stated in the Methods. Please see page 14 line 269-272 and 280-282; page 15 line 297-300.

Question: *Confirming the concordance of NP characteristics in serum and PE by structural finger-prints appeared to be difficult due to the much lower NP concentration in serum. To establish, therefore, that structural fingerprint findings of Fe and O in serum indicate environmental origin requires stronger evidence. Crystallinity may not separate biogenic from abiogenic Fe-oxides.*

Answer: Thanks for your comment. We agree that it is very difficult to distinguish the biogenic from abiogenic Fe-oxides in serum due to their much lower NP concentration. In this revision, we have added two additional evidences to support the external origins of Fe-oxide particles in serum samples:

- 1) As shown in Fig. 3a-f and Supplementary Fig. 7-8, the particles display much larger particle sizes (up to ~150 nm) than endogenous ferritin-derived NPs;
- 2) Furthermore, as shown in Supplementary Fig. 7, the particles bear fused interlocking surface crystallites with the varying orientations of the individual crystallite faces, which is also observed with the particles found in the PE (please see Supplementary Fig. 3) and regarded as typical of high-temperature formation and subsequent crystallization upon

rapid cooling and/or oxidation (*P. Ntl. Acad. Sci. USA* 2016, *113*, 10797-10801; *Environ. Res.* 2019, *176*, 108567).

Therefore, taking together the evidences mentioned above, it is most likely that the magnetite NPs found in serum and PE are of same origin from ambient PM. The additional evidences for magnetite NPs in serum as well as more results for NPs in PE are given in page 8 line 151-155; page 9 line 182-184; and Supplementary Fig. 3 and 7-8.

Question: *It has to be considered that under inflammatory conditions in the lung (e.g., pneumonia, COPD) the epithelial integrity (tight junctions) is compromised and transfer of solutes and NPs occurs in both directions. Biotransformation processes of NPs in the lung and tissues are also to be expected. Can singular findings in a PE sample of one patient (P12) and a serum sample of one other patient (P32) (lines 170/171) be generalized as evidence that these are counterparts in serum and PE?*

Answer: Thanks very much for your suggestion. Firstly, the exchange of solutes and particles between the PE and blood circulation under inflammatory conditions has been clearly stated in the revised manuscript. Please see page 4 line 75-77. The possible biotransformations of NPs in the lung and tissues have also been mentioned in page 6 line 103-104.

Secondly, with regard to the NPs you mentioned, we have added more results from more subjects (P12, P15, P21, and P30 for PE and S32, S19, and S21 for serum) to show their similarity. Please see Supplementary Fig. 9 in the SI.

Question: *The assumption expressed in this manuscript that results of the authors' in vitro RAW studies show the true pro-inflammatory activity of the particles in the human body is an overstatement. The exposure concentration of the cell medium – used as metric - is not equivalent to the actual dose to a cell; also, using one concentration only will not allow to establish dose-effect relationships.*

Answer: We agree that the toxicological results in the previous version of this paper are raw with only a single cell type in a single assay. Therefore, after careful consideration and taking into account the reviewers' comments, we have removed the toxicological assay from the revised manuscript based on the following reasons:

- i) Due to the NP extraction process used in the present study, the surface properties (e.g., surface chemistry, redox potential, and solubility) of NPs might have been altered compared with those in the human body, which may greatly affect the toxicity of the NPs;
- ii) Moreover, the main objective of this study is to probe and characterize exogenous UFPs in the human serum and PE, and, therefore, the toxicological effect of the NPs is likely beyond the scope of the present study.

Therefore, to make the present paper more reasonable, we think that it should be better to be

more concentrated on the characterization of NPs in the PE and serum. Anyway, if you think that the toxicological assay should be kept in this paper, we will be happy to add it into the paper again with proper revisions.

Response to Reviewer #3:

General Comments:

The authors have identified an important aspect of pollution uptake into systemic circulation derived from ultra-fine particulate matter that is not typically addressed with air pollution standards of PM_{2.5}. The critical assessment of the ultra-fine particulate NPs involving chemical, structural and size-related characteristics has been addressed and shown for serum and pleural effusion.

Major Claims: *Linking the sources of ultra-fine particulates such as combustion-derived or traffic related emission NPs to the types of NPs found in serum and PE was succinctly performed and is comparing the liquid-containing nanoparticle host “serum/PE” versus previously published tissue-containing hosts (brain, liver, spleen etc.). Identifying NP in tissue and comparing with exogenous source materials has been widely published and the current work focuses on systemic NPs and their origins. Nanoparticle uptake into blood has been studied previously, but PE as a host of NP is innovative and was detailed in the manuscript.*

Understanding the role of NP in systemic circulation and their origins will be of importance to nanotoxicology and risk assessment in particular and environmental pollution-derived disease developments and public health in general.

Novelty Aspects: *The authors addressed various sources like coal combustion fly ash and automotive exhaust catalyst particles as exogenous matter that translocated to PE via serum uptake using particle shapes, isotopic signatures of Fe and metal concentrations in PE. One important area that was overlooked or not addressed is the dynamic nature of NPs and if this will also be the case for serum and PE. It should be considered whether the translocated particles will be subject to bio-transformations including partial dissolution, reformation to secondary particles (in vivo formation). This should at least be pointed out in the discussion part.*

Study conclusions: *The overall conclusions are well in line with previous studies that investigated the in vivo translocation and toxicity aspects of nanoparticles after exposure. However, the uptake of NPs into PE from systemic circulation has a significant novelty aspect because it relates to nanoparticles in liquid medium instead of tissue interactions and uptake. The authors demonstrate using HAADF STEM and EELS that the chemical and structural nature of the NPs can be linked to exogenous matter. The Fe isotope analyses of PE containing nanoparticles and serum is also novel and allows to pinpoint to the origins of the*

ultra-fine particulate matter and this aspect is original, innovative and deserves publication.

Review Summary: *The manuscript is original but requires several modifications and they are itemized in detail under “Specific Comments” below:*

Specific Comments:

Question: *Line 38:only in recent decades did it become one of the leading global health risks due to dramatically increased anthropogenic sources. Need Reference(s).*

Answer: The reference has been added in the revised manuscript. Please see Ref. 1.

Question: *Line 42:More than 90% of the global population are living in polluted air. Need reference for population data.*

Answer: The population data has been added. Please see page 3 line 41.

Question: *Line 44: ...Long-term exposure to PM_{2.5} is thought to increase mortality and morbidity and shorten life expectancy by causing cardiovascular and respiratory diseases, such as respiratory infections, chronic obstructive pulmonary (COPD), heart attack, and lung cancer. Reference is missing.*

Answer: The references have been added. Please see Ref. 4-5.

Question: *Line 66: The NPs in serum can directly reflect the internal exposure to UFPs. This statement needs to be reworded since it suggests that a direct observation of NP level in serum and dose exposure effects are already known or accepted.*

Answer: Thanks for your suggestion. This sentence has been revised to “At first, we attempted to investigate NPs in serum to directly reflect the internal exposure to UFPs.” Please see page 4 line 68-69.

Question: *Line 72: ...The extracted NPs were purified by enzymatic hydrolysis. Need to clarify that the NPs were extracted from PE.*

Answer: Revision has been made as you suggested. Please see page 5 line 80.

Question: *Line 74:multi-fingerprints (including elemental, structural, and natural isotopic fingerprints) Not sure why the authors refer to “natural isotopic” - if all NPs trapped in PE are characterized, then there is no distinction needed here for “natural”.*

Answer: The “natural isotopic” means natural stable isotopic. To avoid misunderstanding, throughout the manuscript, “natural isotopic” has been changed to “stable isotopic”.

Question: *Line 75:It should be noted that PE samples can only be clinically collected from patients with some specific diseases. So, in this study, both patients and healthy subjects were recruited, and the serum samples were collected from both healthy subjects and patients to compare the NP levels in the human body between healthy subjects and patients. This is contradicting the earlier explanations that PE samples would be used to extract NPs. If PE*

samples are only taken from patients and not for healthy subjects then there is only a comparison possible for serum. Specify.

Answer: The description of the participants has been revised to be clearer. The serum samples were collected from both healthy subjects and patients but the PE samples were only from patients. Thus, the comparison between healthy subjects and patients is indeed only possible with serum samples (please see Supplementary Fig. 1). This point has been clarified in the revised manuscript. Please see page 5 line 84-87.

Question: *Line 88: ... Concentration and size of NPs in human serum. As shown in Fig. 1a, nanoparticle tracking analysis shows that NPs are ubiquitous in all of the human serum samples with the concentration ranging from 1.4×10^8 to 1.0×10^{10} particles mL^{-1} (mean value 2.6×10^9 particles mL^{-1}). The size distribution shows large polydispersity and individual differences within a few to hundreds of nanometers. Such a particle size range actually exceeds the commonly supposed threshold for atmospheric particulate matter to penetrate through the pulmonary alveoli ($< 0.1 \mu\text{m}$), suggesting that the penetrability of particulate matter may be stronger than previously thought.*

1) The statement that NPs are ubiquitous in all of the human serum samples suggests that NPs were classified as coming from different sources. This needs to be restated since the authors do not have this information.

2) Concentration ranges for NPs are strongly dependent on the potential agglomeration tendencies of the NP in serum. Furthermore, the separation of NP will also affect dispersion, agglomeration which can change the concentration ranges for particles in serum.

3) The size distribution of NP in serum depends on agglomeration tendencies since agglomerated particles will be measured as a “larger particle”. This needs to be discussed here when talking about “polydispersity” and size differences.

4) If particles agglomerate and deagglomerate in serum (and during the extraction procedures) then the statement that penetrability of particulate matter may be stronger than previously thought is not proven. The complexity of particle agglomeration tendency needs to be considered here. Also, the authors did not consider that NP can partially dissolve which can reduce size and other factors.

Answer: Thanks very much for your valuable suggestions. According to your suggestions, we have made such revisions:

1) The related statement has been changed to “NPs are widely present in all of the human serum samples ...” to avoid misunderstanding. Please see page 5 line 92.

2) We have taken into account the agglomeration and deagglomeration of NPs when interpreting the results of particle concentration and size measurements all through the revised manuscript. For example, when describing the NTA results of NPs in serum, the

potential effects of NP agglomeration/deagglomeration have been clearly discussed. We have also alerted the readers that the particle size distributions obtained here may not exactly reflect those in ambient environment. Please see page 6 line 101-105. Furthermore, in the Discussion, the effect of NP agglomeration has also been mentioned. Please see page 12 line 238-239. In the caption of Supplementary Fig. 5, the potential effect of NP agglomeration on particle size is also mentioned.

- 3) As responded above, when talking about particle size distribution, the potential effects of NP agglomeration/deagglomeration on particle size have been discussed.
- 4) We have thoroughly revised the manuscript to avoid any unproven premature statements. In this revision, the discussion on the penetrability of UFPs has been moved to the Discussion, and the related statement has been revised to “Considering the potential agglomeration tendency of NPs in body fluids (and in extraction process), it may be premature to conclude that the penetrability of ambient particulate matter is stronger than previously thought. However, our results call for more comprehensive studies on size-dependent health effect of particulate exposure.” Please see page 12 line 232-242. Furthermore, the effect of partial dissolution of NPs has also been mentioned. Please see page 6 line 102.

Question: *Line 98: ...(ii) the NPs in PE are less polydisperse than those in serum (nearly all of the NPs in PE are constrained within 50-200 nm), suggesting that the pleura may have a sieving effect for NPs due probably to the deposition of some large particles at the vascular walls. Again, this is an unproven statement by the authors: Deagglomeration potential has not been addressed; Partial Solubility of NP particles as a result of residence time in PE has not been addressed; NP transformations in PE have not been addressed as a potential factor to affect a smaller particle size range in PE. Also, there would be an “accumulation effect” where NP are either stored in PE or removed from PE.*

Answer: According to your suggestion, the explanation for the less polydispersity of the NPs in PE than in serum has been updated in the revised manuscript, which may be related to the dynamics of NPs (including NP agglomeration/deagglomeration, dissolution, and transformations) and the “accumulation effect” when NPs enter or drain from PE. Please see page 6 line 115-117.

Question: *Line 105: Despite that, we find that the PE-derived NPs can cause significant pro-inflammatory effects at the real doses as in the human body after excluding the effect of endotoxin for all PE samples (Fig. 1e and Supplementary Fig. 3), suggesting that the NPs in the human body have significant health risks. This paragraph is completely confusing. It is not clear how the authors obtain the information on pro-inflammatory effects at “real doses”?*

Need to elaborate. In introduction it was stated that extracted NPs were digested to eliminate protein coatings on the particle surfaces etc which affects the NP reactivity and their in vivo toxicity. To suggest that NPs have significant health risks is generally known and documented, but the actual pro-inflammatory effects of NPs in PE depend on a series of factors which are not addressed by the authors.

Answer: Thanks very much for your comment. After careful consideration and taking into account the reviewers' comments, we have removed the toxicological assay from the revised manuscript based on the following reasons:

- i) The toxicological results are very rough with only a single cell type in a single assay, which is difficult to reflect the real doses and toxic effects of particles in the human body;
- ii) Due to the NP extraction process used in the present study, the surface properties (e.g., surface chemistry, redox potential, and solubility) of NPs might have been altered compared with those in the human body (as you mentioned in the next question), which may greatly affect the toxicity of the NPs;
- iii) Moreover, the main objective of this study is to probe and characterize exogenous UFPs in the human serum and PE, and, therefore, the toxicological effect of the NPs is likely beyond the scope of the present study.

Therefore, to make the present paper more reasonable, we think that it should be better to be more concentrated on the characterization of NPs in the PE and serum. Anyway, if you think that the toxicological assay should be kept in this paper, we will be happy to add it into the paper again with proper revisions.

Question: *Line 111:.... Thus, the toxic effect of the NPs may be more relevant to their chemical nature. The chemical nature of the NP in PE and serum are very divers and to suggest that the toxic effects of the NPs may be more relevant to their chemical nature is far too vague.*

- 1) *Surface area effects of NPs are not addressed;*
- 2) *Redox Potential of NPs is not addressed;*
- 3) *Solubility effects are not addressed.*

Answer: Thanks for your comment. As responded above, the part of toxicological assay has been removed from the revised manuscript.

Question: *Line 113:.... chemical multi-fingerprints: What exactly do the authors mean by “multi-fingerprints”? Extreme diversity in chemical fingerprints?*

Answer: In line 124, “chemical multi-fingerprints” has been specified to “elemental fingerprints”. The chemical multi-fingerprints mean the combination of elemental, structural, and stable isotopic fingerprints, which are defined in page 5 line 82.

Question: *Line 114: Besides the high-abundance elements in organisms. Are the authors suggesting that NPs in PE and serum are composed of elements that are constituents of “high-abundance elements in organisms”? The analyses should focus on the actual NPs extracted from PE and serum. In Figure 1f If the elemental concentrations of Fe >> Mn > Ti > Ba > Sr would reflect elements that are common constituents. The question that needs to be addressed is: What kind of NPs are high in Fe (oxides, phosphates, oxyhydroxides; sulfates etc..) and are the Fe particles magnetic (Fe₃O₄)?and what kind of NPs contain Mn, Ba, Sr, Ti etc. Are these NPs oxides or some other compounds?*

Answer: The confusing statement “Besides the high-abundance elements in organisms” has been removed from the revised manuscript, which actually means that the high-abundance elements in organisms (e.g., C, H, O, N, S) are not included in the present study due to the limitations of the techniques used and thus only the other elements are investigated. Please see page 7 line 125. Furthermore, since the ICP-MS only gives elemental concentrations, the elemental compositions relative to different kinds of NPs are further uncovered by HAADF-STEM with EDXS and EELS analysis.

Question: *Line 117: ... Note that Al, Fe, Ti, Mn, Ba, and Sr; which are rock-forming elements or relatively abundant elements in the Earth’s crust, show relatively high concentrations in the NPs (Fig. 1f). Such elemental fingerprints suggest that the NPs are of external abiogenic sources. This is not correct! Fe (iron) can form biomineralized ferritin NPs from endogenous iron source and the presence of Fe in serum and PE could be derived from either endogenous or exogenous or both sources. This needs to be clarified in the text.*

Answer: The wrong statement on Fe has been corrected. Please see page 7 line 127. The possibility of forming endogenous magnetite NPs from biomineralization of ferritin in the human body has also been clearly clarified in page 8 line 149-151.

Question: *Line 120:Remarkably, Pt, a characteristic element in vehicle exhaust catalysts, is found in some samples from benign patients (with no application of Pt-containing agents), suggesting that some NPs may originate from vehicle exhausts. Pt is also used in many other catalyst applications besides automotive. I would recommend not to make extrapolations in the Results section.*

Answer: Thanks for your suggestion. In this revision, the extrapolations of the Pt sources have been removed from the Results. Please see page 7 line 130-131.

Question: *Line 124: The elemental fingerprints of NPs also show large individual differences (Fig. 1g), which can partly explain the toxic effect of the NPs. I think this sentence needs to be reworded. Figures 1e and g together indicate how NPs for different samples (P) show elemental tendencies that are linked to an inflammatory response: For example P34 and P36 in Figure 1g have high Vanadium which is indicated in Figure 1e to correspond to*

an elevated inflammatory response.

Answer: As responded above, the content of toxicological assay has been removed from the revised manuscript. However, we thank you for your valuable suggestion which will be taken into account in our future studies.

Question: *Line 134: ... Abundant Fe-bearing NPs are found, as Fe makes up 0.17%-12.27% of the mass of the NPs. Need to refer to the structures of the Fe-NPs as observed in HAADF STEM. 1) A number of spherical NPs consisting of Fe and O are found (section I in Fig. 2a; Fig. 2b-d). Indexing of the lattice fringes and the FFT pattern of the particle are consistent with the magnetite crystal structure (Fig. 2e-f).*

Answer: The Fe mass percentage of NPs is introduced with reference to the results obtained in HAADF-STEM. Please see page 7-8 line 143-147.

Question: *Line 129: Structural fingerprints of NPs in PE. The structural information is gained from the HAADF-STEM (Morphological; Crystallographic). The EDS and EELS data is chemical and electrochemical information and the heading "Structural Fingerprints" needs to be expanded.*

Answer: The subtitle has been revised to "Structural fingerprints with enhanced chemical identities of NPs in PE" to more properly summarize the content of this section. Please see page 7 line 135.

Question: *Line 140: Recall that endogenous magnetite particles formed via in situ crystallization within the 8-nm-diameter core of ferritin should have a euhedral angular crystal morphology. Biomineralized Ferritin NPs that form in the ferritin nanocage are actually not angular and are composed predominantly of iron oxyhydroxide nanoparticles, This needs to be corrected. Only ferritins that form in vivo should be considered and not iron oxide nanoparticles that are synthesized using a ferritin precursor nanocage.*

Answer: Thanks for your suggestion. The related statement has been corrected to "endogenous magnetite particles can form via in vivo crystallization within the 8-nm-diameter core of ferritin" to avoid misunderstanding. Please see page 8 line 150-151.

Question: *Line 142: ... Thus, the magnetite NPs observed here contrast with the biogenic ones but resemble the pollution NPs found in the human brain reported previously. Several of the brain -containing magnetite from reference 4 also have spherical and angular morphologies. The main difference between the ferritin derived iron oxide NPs and those found in the brain study were the larger particle sizes of the magnetite! It would be good if the authors refer to the particle size of the magnetite found in the PE and serum samples. The example in Figure 2 shows spherical magnetite NPs.*

Answer: Thanks for your valuable suggestion. In this revision, we have emphasized the

difference in particle size between the magnetite NPs found in PE and serum and endogenous ferritin-derived NPs. This indeed provides a strong evidence for the external origins of the magnetite NPs. Please see page 8 line 151-152; page 9 line 184; and Supplementary Fig. 7-8.

Question: *Line 146:....Besides magnetite, we also observe other Fe-bearing NPs such as rounded crystal NPs consisting of Fe, Mn, and Ni (Fig. 2h-j and Supplementary Fig. 5) that resemble Fe-Mn-Ni alloy particles existing in PM_{2.5} emitted from ferroalloy plants. Are these oxides or metal spheres? This is not clear in the text or in Figure 2. Fe, Mn, Ni are very common in fly ash from coal combustion plants. They typically occur as “fly ash spheres” with a high percentage of transition metal oxides that are contained in a glassy (Si, Al) matrix. To be derived from a ferroalloy plant the NP would be reduced metals or metallic. If the authors relate the Fe-Mn-Ni NPs to ferroalloy plants they need to distinguish them from fly ash spheres.*

Answer: Thanks for your question. From EDX, we found that the FeMnNi NPs obtained here only consist of Fe, Mn, and Ni, and no Si or Al which are common components of fly ash are found. Thus, it is more likely that these particles are Fe-Mn-Ni alloy particles existing in PM emitted from ferroalloy plants rather than from fly ash particles. This result has been clarified in the revised manuscript. Please see page 8 line 160 and Supplementary Fig. 4.

Question: *Line 151:.... Elemental Cu is scarcely present in natural environment or human body, and it is most likely to be emitted by electric motors in indoor environments. Cu-solution is typically used as a wood preservative and has been shown to form nanoparticles inside the lumber. Any lumber cutting or manufacturing process could set free NPs and the authors need to consider this source as well.*

Answer: Thanks for your useful suggestion. The wood preservative as another potential source of Cu NPs has been added to the revised manuscript. Please see page 8 line 165-166.

Question: *Line 155: We also find hexagonal Ti-bearing particles (Fig. 2p). Lattice indexing and EDXS mapping indicate that they are crystal TiO₂ NPs (Fig. 2q-s), which are one of the most produced engineered nanomaterials. Should also mention that industrially produced nano-TiO₂ often is spherical in nature or occurs as nanorods which have a hexagonal crystal lattice.*

Answer: The knowledge you mentioned has been added to the revised manuscript. Please see page 9 line 170-171.

Question: *Line 169:... In addition, we identified a type of special fusiform NPs with a mineral-like elemental composition (C, O, Ca, and P). Not sure what fusiform NPs means. May need to elaborate. The composition C, O, Ca and P may be suggestive of a) Calcium Carbonate and Calcium Phosphate which can have both exogenic or endogenic origins.*

Answer: Firstly, the “fusiform NPs” has been elaborated to “spindle-shaped NPs with a mineral-like elemental composition”. Please see page 9 line 186. Furthermore, following your suggestion, we have clarified that “The elemental composition may be suggestive of calcium carbonate or calcium phosphate which can have both exogenic or endogenic origins”. Please see page 9-10 line 188-189.

Question: *Line 206:… The chemical multi-fingerprinting shows that the NPs mainly originate from external particulate pollution, particularly, combustion-related particulate emission. I would suggest to include “structural” fingerprinting since fly ash spheres can be identified and this would suggest some combustion sources.*

Answer: In page 11 line 225, “chemical multi-fingerprinting” has been specified to “elemental, structural, and stable isotopic fingerprints”.

Question: *Line 217:…. which also needs to take into account the effect of protein corona on the NPs in the future studies And also needs to take into account solubility of NPs from combustion sources after uptake into serum and PE.*

Answer: The “solubility after uptake into human body” of NPs has been mentioned in the revised manuscript. Please see page 12 line 249.

Question: *Line 218: … The occurrence states of exogenous NPs in more human organs need to be studied to fully understand the dynamics and life cycle of ambient UFPs in the human body; Also, the authors should mention that NPs in PE and serum will have a residence time which will affect particle transformations and solubility and clearance.*

Answer: We have mentioned that the NPs in serum and PE will have a residence time which can affect particle transformation, solubility and clearance in the Discussion. Please see page 12 line 251-252.

Finally, we thank again the editor and the reviewers for your great efforts on improving the quality of this manuscript. We are looking forward to hearing your decision soon.

Thank you very much!

Best wishes,

Yours sincerely,

Dr. Qian Liu and Dr. Guibin Jiang

Reviewers' Comments:

Reviewer #1:

Remarks to the Author:

While the line numbers in the rebuttal do not match those in the document, all changes indicated have been made, and they address my comments well. I should note that in response to Q8, I have no objection to the reporting of the in vitro toxicological assay per se – I am happy for this data to be included, as long as the limitations are emphasised (e.g. single cell type, single dose, presence of endotoxin, concentration employed is not necessarily 'real', extraction may alter properties of recovered particles, etc).

Reviewer #2:

Remarks to the Author:

The manuscript is significantly improved. However, there is still the problem with asserting that nanostructures seen in serum are evidence of inhaled ambient ultrafine particles which translocated into the blood circulation, despite the authors' statement that "NP levels in serum were too low to perform a comprehensive characterization" (line 71).

That leaves the NPs in PE samples as evidence of circulatory translocation for which, however, no persuasive evidence is offered: NPs in PE for the most part represent inhaled ambient NPs that translocated interstitially to the pleura; but this does not provide convincing evidence that these NPs originated from the blood circulation. For a minority of them, though, their presence in PE might be viewed as indicating circulatory origin which - although consistent with inhaled NPs having translocated into the blood circulation - is not compelling from the results of this study without further proof.

The authors' hypothesis "that the UFPs in the PE and circulatory system should keep homology and thus PE can serve as a host of inhaled UFPs in the human body" (lines 79-81) cannot be verified by the present study design. In order to demonstrate this requires additional research aimed at differentiating the authors' suggested circulation derived NPs from the interstitially translocated NPs in PE. At best, the authors may propose that inhaled NPs entering the parietal lymphatic stomata will reach the venous circulation via the Thoracic Lymph Duct and then, through the arterial circulation, extravasate at the pleura. An interesting hypothesis, but hard to prove in vivo. Thus, the statement that this study provides "clear evidence" in humans for translocation into circulation and for systemic health effects of ambient UFPs is overstated (lines 31; 237). Clear evidence of such translocation for the first time in humans has been provided earlier for NPs (Miller et al, 2017; ref17) and most recently for UFPs (Calderon-Garciduenas et al, 2019; ref 18). This needs to be emphasized more strongly (lines 57 - 59).

Reviewer #3:

Remarks to the Author:

After carefully reviewing the edited manuscript and going over the "Response to Referees" section, I have only one additional comment/suggestion that is highlighted in yellow/green under the reviewer #3 portion. Please also see the attached file for details.

Question: Line 66: The NPs in serum can directly reflect the internal exposure to UFPs. This statement needs to be reworded since it suggests that a direct observation of NP level in serum and dose exposure effects are already known or accepted.

Answer: Thanks for your suggestion. This sentence has been revised to "At first, we attempted to investigate NPs in serum to directly reflect the internal exposure to UFPs." Please see page 4 line 68-69. I would recommend to add an additional sentence here to state: " Internal exposure of NPs

in serum does not provide information on any dose-exposure relation

Response to Reviewer #3:

General Comments:

The authors have identified an important aspect of pollution uptake into systemic circulation derived from ultra-fine particulate matter that is not typically addressed with air pollution standards of PM_{2.5}. The critical assessment of the ultra-fine particulate NPs involving chemical, structural and size-related characteristics has been addressed and shown for serum and pleural effusion.

Major Claims: *Linking the sources of ultra-fine particulates such as combustion-derived or traffic related emission NPs to the types of NPs found in serum and PE was succinctly performed and is comparing the liquid-containing nanoparticle host “serum/PE” versus previously published tissue-containing hosts (brain, liver, spleen etc.). Identifying NP in tissue and comparing with exogenous source materials has been widely published and the current work focuses on systemic NPs and their origins. Nanoparticle uptake into blood has been studied previously, but PE as a host of NP is innovative and was detailed in the manuscript. Understanding the role of NP in systemic circulation and their origins will be of importance to nanotoxicology and risk assessment in particular and environmental pollution-derived disease developments and public health in general.*

Novelty Aspects: *The authors addressed various sources like coal combustion fly ash and automotive exhaust catalyst particles as exogenous matter that translocated to PE via serum uptake using particle shapes, isotopic signatures of Fe and metal concentrations in PE. One important area that was overlooked or not addressed is the dynamic nature of NPs and if this will also be the case for serum and PE. It should be considered whether the translocated particles will be subject to bio-transformations including partial dissolution, reformation to secondary particles (in vivo formation). This should at least be pointed out in the discussion part.*

Study conclusions: *The overall conclusions are well in line with previous studies that investigated the in vivo translocation and toxicity aspects of nanoparticles after exposure. However, the uptake of NPs into PE from systemic circulation has a significant novelty aspect because it relates to nanoparticles in liquid medium instead of tissue interactions and uptake. The authors demonstrate using HAADF STEM and EELS that the chemical and structural nature of the NPs can be linked to exogenous matter. The Fe isotope analyses of PE containing nanoparticles and serum is also novel and allows to pinpoint to the origins of the ultra-fine particulate matter and this aspect is original, innovative and deserves publication.*

Review Summary: *The manuscript is original but requires several modifications and they are itemized in detail under “Specific Comments” below:*

Specific Comments:

Question: Line 38:only in recent decades did it become one of the leading global health risks due to dramatically increased anthropogenic sources. Need Reference(s).

Answer: The reference has been added in the revised manuscript. Please see Ref. 1.

Question: Line 42:More than 90% of the global population are living in polluted air. Need reference for population data.

Answer: The population data has been added. Please see page 3 line 41.

Question: Line 44: ...Long-term exposure to PM_{2.5} is thought to increase mortality and morbidity and shorten life expectancy by causing cardiovascular and respiratory diseases, such as respiratory infections, chronic obstructive pulmonary (COPD), heart attack, and lung cancer. Reference is missing.

Answer: The references have been added. Please see Ref. 4-5.

Question: Line 66: The NPs in serum can directly reflect the internal exposure to UFPs. This statement needs to be reworded since it suggests that a direct observation of NP level in serum and dose exposure effects are already known or accepted.

Answer: Thanks for your suggestion. This sentence has been revised to “At first, we attempted to investigate NPs in serum to directly reflect the internal exposure to UFPs.” Please see page 4 line 68-69. I would recommend to add an additional sentence here to state: “Internal exposure of NPs in serum does not provide information on any dose-exposure relationship”.

Question: Line 72: ...The extracted NPs were purified by enzymatic hydrolysis Need to clarify that the NPs were extracted from PE.

Answer: Revision has been made as you suggested. Please see page 5 line 80.

Question: Line 74:multi-fingerprints (including elemental, structural, and natural isotopic fingerprints) Not sure why the authors refer to “natural isotopic” - if all NPs trapped in PE are characterized, then there is no distinction needed here for “natural”.

Answer: The “natural isotopic” means natural stable isotopic. To avoid misunderstanding, throughout the manuscript, “natural isotopic” has been changed to “stable isotopic”.

Question: Line 75:It should be noted that PE samples can only be clinically collected from patients with some specific diseases. So, in this study, both patients and healthy subjects were recruited, and the serum samples were collected from both healthy subjects and patients to compare the NP levels in the human body between healthy subjects and patients. This is contradicting the earlier explanations that PE samples would be used to extract NPs. If PE samples are only taken from patients and not for healthy subjects then there is only a comparison possible for serum. Specify.

Answer: The description of the participants has been revised to be clearer. The serum samples

were collected from both healthy subjects and patients but the PE samples were only from patients. Thus, the comparison between healthy subjects and patients is indeed only possible with serum samples (please see Supplementary Fig. 1). This point has been clarified in the revised manuscript. Please see page 5 line 84-87.

Question: *Line 88:... Concentration and size of NPs in human serum. As shown in Fig. 1a, nanoparticle tracking analysis shows that NPs are ubiquitous in all of the human serum samples with the concentration ranging from 1.4×10^8 to 1.0×10^{10} particles mL^{-1} (mean value 2.6×10^9 particles mL^{-1}). The size distribution shows large polydispersity and individual differences within a few to hundreds of nanometers. Such a particle size range actually exceeds the commonly supposed threshold for atmospheric particulate matter to penetrate through the pulmonary alveoli ($< 0.1 \mu m$), suggesting that the penetrability of particulate matter may be stronger than previously thought.*

1) *The statement that NPs are ubiquitous in all of the human serum samples suggests that NPs were classified as coming from different sources. This needs to be restated since the authors do not have this information.*

2) *Concentration ranges for NPs are strongly dependent on the potential agglomeration tendencies of the NP in serum. Furthermore, the separation of NP will also affect dispersion, agglomeration which can change the concentration ranges for particles in serum.*

3) *The size distribution of NP in serum depends on agglomeration tendencies since agglomerated particles will be measured as a “larger particle”. This needs to be discussed here when talking about “polydispersity” and size differences.*

4) *If particles agglomerate and deagglomerate in serum (and during the extraction procedures) then the statement that penetrability of particulate matter may be stronger than previously thought is not proven. The complexity of particle agglomeration tendency needs to be considered here. Also, the authors did not consider that NP can partially dissolve which can reduce size and other factors.*

Answer: Thanks very much for your valuable suggestions. According to your suggestions, we have made such revisions:

1) The related statement has been changed to “NPs are widely present in all of the human serum samples ...” to avoid misunderstanding. Please see page 5 line 92.

2) We have taken into account the agglomeration and deagglomeration of NPs when interpreting the results of particle concentration and size measurements all through the revised manuscript. For example, when describing the NTA results of NPs in serum, the potential effects of NP agglomeration/deagglomeration have been clearly discussed. We have also altered the readers that the particle size distributions obtained here may not exactly reflect those in ambient environment. Please see page 6 line 101-105. Furthermore,

in the Discussion, the effect of NP agglomeration has also been mentioned. Please see page 12 line 238-239. In the caption of Supplementary Fig. 5, the potential effect of NP agglomeration on particle size is also mentioned.

- 3) As responded above, when talking about particle size distribution, the potential effects of NP agglomeration/deagglomeration on particle size have been discussed.
- 4) We have thoroughly revised the manuscript to avoid any unproven premature statements. In this revision, the discussion on the penetrability of UFPs has been moved to the Discussion, and the related statement has been revised to “Considering the potential agglomeration tendency of NPs in body fluids (and in extraction process), it may be premature to conclude that the penetrability of ambient particulate matter is stronger than previously thought. However, our results call for more comprehensive studies on size-dependent health effect of particulate exposure.” Please see page 12 line 232-242. Furthermore, the effect of partial dissolution of NPs has also been mentioned. Please see page 6 line 102.

Question: *Line 98: ... (ii) the NPs in PE are less polydisperse than those in serum (nearly all of the NPs in PE are constrained within 50-200 nm), suggesting that the pleura may have a sieving effect for NPs due probably to the deposition of some large particles at the vascular walls. Again, this is an unproven statement by the authors: Deagglomeration potential has not been addressed; Partial Solubility of NP particles as a result of residence time in PE has not been addressed; NP transformations in PE have not been addressed as a potential factor to affect a smaller particle size range in PE. Also, there would be an “accumulation effect” where NP are either stored in PE or removed from PE.*

Answer: According to your suggestion, the explanation for the less polydispersity of the NPs in PE than in serum has been updated in the revised manuscript, which may be related to the dynamics of NPs (including NP agglomeration/deagglomeration, dissolution, and transformations) and the “accumulation effect” when NPs enter or drain from PE. Please see page 6 line 115-117.

Question: *Line 105:.... Despite that, we find that the PE-derived NPs can cause significant pro-inflammatory effects at the real doses as in the human body after excluding the effect of endotoxin for all PE samples (Fig. 1e and Supplementary Fig. 3), suggesting that the NPs in the human body have significant health risks. This paragraph is completely confusing. It is not clear how the authors obtain the information on pro-inflammatory effects at “real doses”? Need to elaborate. In introduction it was stated that extracted NPs were digested to eliminate protein coatings on the particle surfaces etc which affects the NP reactivity and their in vivo toxicity. To suggest that NPs have significant health risks is generally known and documented,*

but the actual pro-inflammatory effects of NPs in PE depend on a series of factors which are not addressed by the authors.

Answer: Thanks very much for your comment. After careful consideration and taking into account the reviewers' comments, we have removed the toxicological assay from the revised manuscript based on the following reasons:

- i) The toxicological results are very rough with only a single cell type in a single assay, which is difficult to reflect the real doses and toxic effects of particles in the human body;
- ii) Due to the NP extraction process used in the present study, the surface properties (e.g., surface chemistry, redox potential, and solubility) of NPs might have been altered compared with those in the human body (as you mentioned in the next question), which may greatly affect the toxicity of the NPs;
- iii) Moreover, the main objective of this study is to probe and characterize exogenous UFPs in the human serum and PE, and, therefore, the toxicological effect of the NPs is likely beyond the scope of the present study.

Therefore, to make the present paper more reasonable, we think that it should be better to be more concentrated on the characterization of NPs in the PE and serum. Anyway, if you think that the toxicological assay should be kept in this paper, we will be happy to add it into the paper again with proper revisions.

Question: *Line 111: Thus, the toxic effect of the NPs may be more relevant to their chemical nature. The chemical nature of the NP in PE and serum are very divers and to suggest that the toxic effects of the NPs may be more relevant to their chemical nature is far too vague.*

- 1) *Surface area effects of NPs are not addressed;*
- 2) *Redox Potential of NPs is not addressed;*
- 3) *Solubility effects are not addressed.*

Answer: Thanks for your comment. As responded above, the part of toxicological assay has been removed from the revised manuscript.

Question: *Line 113:.... chemical multi-fingerprints: What exactly do the authors mean by "multi-fingerprints"? Extreme diversity in chemical fingerprints?*

Answer: In line 124, "chemical multi-fingerprints" has been specified to "elemental fingerprints". The chemical multi-fingerprints mean the combination of elemental, structural, and stable isotopic fingerprints, which are defined in page 5 line 82.

Question: *Line 114:.... Besides the high-abundance elements in organisms. Are the authors suggesting that NPs in PE and serum are composed of elements that are constituents of "high-abundance elements in organisms"? The analyses should focus on the actual NPs extracted from PE and serum. In Figure 1f the elemental concentrations of Fe>> Mn> Ti> Ba > Sr*

would reflect elements that are common constituents. The question that needs to be addressed is: What kind of NPs are high in Fe (oxides, phosphates, oxyhydroxides; sulfates etc..) and are the Fe particles magnetic (Fe_3O_4)?and what kind of NPs contain Mn, Ba, Sr, Ti etc. Are these NPs oxides or some other compounds?

Answer: The confusing statement “Besides the high-abundance elements in organisms” has been removed from the revised manuscript, which actually means that the high-abundance elements in organisms (e.g., C, H, O, N, S) are not included in the present study due to the limitations of the techniques used and thus only the other elements are investigated. Please see page 7 line 125. Furthermore, since the ICP-MS only gives elemental concentrations, the elemental compositions relative to different kinds of NPs are further uncovered by HAADF-STEM with EDXS and EELS analysis.

Question: Line 117: ... Note that Al, Fe, Ti, Mn, Ba, and Sr, which are rock-forming elements or relatively abundant elements in the Earth's crust, show relatively high concentrations in the NPs (Fig. 1f). Such elemental fingerprints suggest that the NPs are of external abiogenic sources. This is not correct! Fe (iron) can form biomineralized ferritin NPs from endogenous iron source and the presence of Fe in serum and PE could be derived from either endogenous or exogenous or both sources. This needs to be clarified in the text.

Answer: The wrong statement on Fe has been corrected. Please see page 7 line 127. The possibility of forming endogenous magnetite NPs from biomineralization of ferritin in the human body has also been clearly clarified in page 8 line 149-151.

Question: Line 120:Remarkably, Pt, a characteristic element in vehicle exhaust catalysts, is found in some samples from benign patients (with no application of Pt-containing agents), suggesting that some NPs may originate from vehicle exhausts. Pt is also used in many other catalyst applications besides automotive. I would recommend not to make extrapolations in the Results section.

Answer: Thanks for your suggestion. In this revision, the extrapolations of the Pt sources have been removed from the Results. Please see page 7 line 130-131.

Question: Line 124: The elemental fingerprints of NPs also show large individual differences (Fig. 1g), which can partly explain the toxic effect of the NPs. I think this sentence needs to be reworded. Figures 1 e and g together indicate how NPs for different samples (P) show elemental tendencies that are linked to an inflammatory response: For example P34 and P36 in Figure 1g have high Vanadium which is indicated in Figure 1e to correspond to an elevated inflammatory response.

Answer: As responded above, the content of toxicological assay has been removed from the revised manuscript. However, we thank you for your valuable suggestion which will be taken into account in our future studies.

Question: *Line 134: ... Abundant Fe-bearing NPs are found, as Fe makes up 0.17%-12.27% of the mass of the NPs. Need to refer to the structures of the Fe-NPs as observed in HAADF STEM. 1) A number of spherical NPs consisting of Fe and O are found (section I in Fig. 2a; Fig. 2b-d). Indexing of the lattice fringes and the FFT pattern of the particle are consistent with the magnetite crystal structure (Fig. 2e-f).*

Answer: The Fe mass percentage of NPs is introduced with reference to the results obtained in HAADF-STEM. Please see page 7-8 line 143-147.

Question: *Line 129: Structural fingerprints of NPs in PE. The structural information is gained from the HAADF-STEM (Morphological; Crystallographic). The EDS and EELS data is chemical and electrochemical information and the heading “Structural Fingerprints” needs to be expanded.*

Answer: The subtitle has been revised to “Structural fingerprints with enhanced chemical identities of NPs in PE” to more properly summarize the content of this section. Please see page 7 line 135.

Question: *Line 140: Recall that endogenous magnetite particles formed via in situ crystallization within the 8-nm-diameter core of ferritin should have a euhedral angular crystal morphology. Biomineralized Ferritin NPs that form in the ferritin nanocage are actually not angular and are composed predominantly of iron oxyhydroxide nanoparticles, This needs to be corrected. Only ferritins that form in vivo should be considered and not iron oxide nanoparticles that are synthesized using a ferritin precursor nanocage.*

Answer: Thanks for your suggestion. The related statement has been corrected to “endogenous magnetite particles can form via in vivo crystallization within the 8-nm-diameter core of ferritin” to avoid misunderstanding. Please see page 8 line 150-151.

Question: *Line 142: ... Thus, the magnetite NPs observed here contrast with the biogenic ones but resemble the pollution NPs found in the human brain reported previously. Several of the brain -containing magnetite from reference 4 also have spherical and angular morphologies. The main difference between the ferritin derived iron oxide NPs and those found in the brain study were the larger particle sizes of the magnetite! It would be good if the authors refer to the particle size of the magnetite found in the PE and serum samples. The example in Figure 2 shows spherical magnetite NPs.*

Answer: Thanks for your valuable suggestion. In this revision, we have emphasized the difference in particle size between the magnetite NPs found in PE and serum and endogenous ferritin-derived NPs. This indeed provides a strong evidence for the external origins of the magnetite NPs. Please see page 8 line 151-152; page 9 line 184; and Supplementary Fig. 7-8.

Question: *Line 146:Besides magnetite, we also observe other Fe-bearing NPs such as*

rounded crystal NPs consisting of Fe, Mn, and Ni (Fig. 2h-j and Supplementary Fig. 5) that resemble Fe-Mn-Ni alloy particles existing in PM_{2.5} emitted from ferroalloy plants. Are these oxides or metal spheres? This is not clear in the text or in Figure 2. Fe, Mn, Ni are very common in fly ash from coal combustion plants. They typically occur as “fly ash spheres” with a high percentage of transition metal oxides that are contained in a glassy (Si, Al) matrix. To be derived from a ferroalloy plant the NP would be reduced metals or metallic. If the authors relate the Fe-Mn-Ni NPs to ferroalloy plants they need to distinguish them from fly ash spheres.

Answer: Thanks for your question. From EDX, we found that the FeMnNi NPs obtained here only consist of Fe, Mn, and Ni, and no Si or Al which are common components of fly ash are found. Thus, it is more likely that these particles are Fe-Mn-Ni alloy particles existing in PM emitted from ferroalloy plants rather than from fly ash particles. This result has been clarified in the revised manuscript. Please see page 8 line 160 and Supplementary Fig. 4.

Question: *Line 151:.... Elemental Cu is scarcely present in natural environment or human body, and it is most likely to be emitted by electric motors in indoor environments. Cu-solution is typically used as a wood preservative and has been shown to form nanoparticles inside the lumber. Any lumber cutting or manufacturing process could set free NPs and the authors need to consider this source as well.*

Answer: Thanks for your useful suggestion. The wood preservative as another potential source of Cu NPs has been added to the revised manuscript. Please see page 8 line 165-166.

Question: *Line 155: We also find hexagonal Ti-bearing particles (Fig. 2p). Lattice indexing and EDXS mapping indicate that they are crystal TiO₂ NPs (Fig. 2q-s), which are one of the most produced engineered nanomaterials. Should also mention that industrially produced nano-TiO₂ often is spherical in nature or occurs as nanorods which have a hexagonal crystal lattice.*

Answer: The knowledge you mentioned has been added to the revised manuscript. Please see page 9 line 170-171.

Question: *Line 169: ... In addition, we identified a type of special fusiform NPs with a mineral-like elemental composition (C, O, Ca, and P). Not sure what fusiform NPs means. May need to elaborate. The composition C, O, Ca and P may be suggestive of a) Calcium Carbonate and Calcium Phosphate which can have both exogenic or endogenic origins.*

Answer: Firstly, the “fusiform NPs” has been elaborated to “spindle-shaped NPs with a mineral-like elemental composition”. Please see page 9 line 186. Furthermore, following your suggestion, we have clarified that “The elemental composition may be suggestive of calcium carbonate or calcium phosphate which can have both exogenic or endogenic origins”. Please see page 9-10 line 188-189.

Question: *Line 206: ... The chemical multi-fingerprinting shows that the NPs mainly originate from external particulate pollution, particularly, combustion-related particulate emission. I would suggest to include “structural” fingerprinting since fly ash spheres can be identified and this would suggest some combustion sources.*

Answer: In page 11 line 225, “chemical multi-fingerprinting” has been specified to “elemental, structural, and stable isotopic fingerprints”.

Question: *Line 217:.... which also needs to take into account the effect of protein corona on the NPs in the future studies And also needs to take into account solubility of NPs from combustion sources after uptake into serum and PE.*

Answer: The “solubility after uptake into human body” of NPs has been mentioned in the revised manuscript. Please see page 12 line 249.

Question: *Line 218: ... The occurrence states of exogenous NPs in more human organs need to be studied to fully understand the dynamics and life cycle of ambient UFPs in the human body; Also, the authors should mention that NPs in PE and serum will have a residence time which will affect particle transformations and solubility and clearance.*

Answer: We have mentioned that the NPs in serum and PE will have a residence time which can affect particle transformation, solubility and clearance in the Discussion. Please see page 12 line 251-252.

Finally, we thank again the editor and the reviewers for your great efforts on improving the quality of this manuscript. We are looking forward to hearing your decision soon.

Thank you very much!

Best wishes,

Yours sincerely,

Dr. Qian Liu and Dr. Guibin Jiang

Response to Reviewer Comments

We really appreciate the reviewers' comments on our revised manuscript entitled "Chemical multi-fingerprinting of exogenous ultrafine particles in human serum and pleural effusion" (NCOMMS-19-21077A). We have now finished the revision again according to reviewers' comments. Each point raised by the reviewers has been carefully considered and addressed beneath. The changes in the manuscript are marked using the "track changes" function. In this response letter, the reviewers' comments are in blue font and the author responses are in black font.

Response to Reviewer #1:

Question: *While the line numbers in the rebuttal do not match those in the document, all changes indicated have been made, and they address my comments well. I should note that in response to Q8, I have no objection to the reporting of the in vitro toxicological assay per se - I am happy for this data to be included, as a long as the limitations are emphasised (e.g. single cell type, single dose, presence of endotoxin, concentration employed is not necessarily "real", extraction may alter properties of recovered particles, etc).*

Answer: Thanks for your suggestion. According to your suggestion, the in vitro toxicological assay has been added back to the revised manuscript. Considering that the other two reviewers all have accepted that the toxicological assay can be removed from the manuscript, we place it in the Supporting information (please see the Supplementary Fig. 3), and the limitations of the assay are emphasized in the caption of Supplementary Fig. 3.

Response to Reviewer #2:

Question: *The manuscript is significantly improved. However, there is still the problem with asserting that nanostructures seen in serum are evidence of inhaled ambient ultrafine particles which translocated into the blood circulation, despite the authors' statement that "NP levels in serum were too low to perform a comprehensive characterization" (line 71).*

That leaves the NPs in PE samples as evidence of circulatory translocation for which, however, no persuasive evidence is offered: NPs in PE for the most part represent inhaled ambient NPs that translocated interstitially to the pleura; but this does not provide convincing evidence that these NPs originated from the blood circulation. For a minority of them, though, their presence in PE might be viewed as indicating circulatory origin which - although consistent with inhaled NPs having translocated into the blood circulation - is not compelling from the results of this study without further proof.

The authors' hypothesis "that the UFPs in the PE and circulatory system should keep homology and thus PE can serve as a host of inhaled UFPs in the human body" (lines 79-81) cannot be verified by the present study design. In order to demonstrate this requires additional research aimed at differentiating the authors' suggested circulation derived NPs from the interstitially translocated NPs in PE. At best, the authors may propose that inhaled NPs entering the parietal lymphatic stomata will reach the venous circulation via the Thoracic Lymph Duct and then, through the arterial circulation, extravasate at the pleura. An interesting hypothesis, but hard to prove in vivo.

Thus, the statement that this study provides "clear evidence" in humans for translocation into circulation and for systemic health effects of ambient UFPs is overstated (lines 31; 237). Clear evidence of such translocation for the first time in humans has been provided earlier for NPs (Miller et al, 2017; ref 17) and most recently for UFPs (Calderon-Garciduenas et al, 2019; ref 18). This needs to be emphasized more strongly (lines 57-59).

Answer: Thanks for your valuable suggestions. First, we have further mitigated the conclusion in this paper to avoid overstatement. For example, the statement "provide a clear evidence" has been changed to "provide evidence" in line 31 and 239 to make it more balanced.

Second, we have further narrowed the hypothesis that "the UFPs in the PE and circulatory system should keep homology and PE can serve as a host of inhaled UFPs in the human body" to "PE can serve as a potential host of inhaled UFPs in the human body". In this revision, we focus more on the use of PE-derived NPs to reflect the inhaled UFPs, and avoid prematurely claiming that the NPs in the PE may come from the blood circulation.

Third, according to your suggestion, we have emphasized more strongly the recent works by Miller et al. and Calderón-Garcidueñas et al. in the Introduction. Please see page 4 line 58-62.

Response to Reviewer #3:

Question: *After carefully reviewing the edited manuscript and going over the "Response to Referees" section, I have only one additional comment/suggestion that is highlighted in yellow/green under the reviewer #3 portion. Please also see the attached file for details.*

Question: *Line 66: The NPs in serum can directly reflect the internal exposure to UFPs. This statement needs to be reworded since it suggests that a direct observation of NP level in serum and dose exposure effects are already known or accepted.*

Answer: *Thanks for your suggestion. This sentence has been revised to "At first, we attempted to investigate NPs in serum to directly reflect the internal exposure to UFPs." Please see page 4 line 68-69. I would recommend to add an additional sentence here to state: "Internal exposure of NPs in serum does not provide information on any dose-exposure*

relationship".

Answer: Thanks for your suggestion. The sentence "*Internal exposure of NPs in serum does not provide information on any dose-exposure relationship*" has been added in the revised paper. Please see page 4 line 74-75.

Finally, we thank again the editor and reviewers for your great efforts on improving the quality of this manuscript. We are looking forward to hearing your decision soon.

Thank you very much!

Best wishes,

Yours sincerely,

Dr. Qian Liu and Dr. Guibin Jiang